# The South American MicroBiome Archive (saMBA): enriching the microbiome field by studying neglected populations

Benjamin Valderrama [1,2] ✉, Paulina Calderón-Romero[3,4], Thomaz F. S. Bastiaanssen[5], Aonghus Lavelle[1,2], Gerard Clarke [1,6] & John F. Cryan [1,2] ✉

The human gut microbiome is associated with numerous health outcomes, often in a region-specific manner. Unfortunately, global microbiome research remains profoundly imbalanced: over 70% of sequenced human microbiomes originate from Europe and North America, which together represent only 15% of the world's population. To address this disparity, we developed saMBA—the largest archive of gut microbiomes from South America, one of the world's most microbiome-diverse regions but also among the least studied. The archive comprises 33 studies, ~73% of which had not been included in any previous compendium. A total of 3382 samples were reanalysed, of which 2913 were successfully included after applying quality filters. By leveraging this resource, we reveal both high within-population diversity and between-population uniqueness in the continent, expanding our current understanding of the gut microbiome to be more globally representative. Additionally, saMBA reveals that much of the region's gut microbiome diversity remains under-characterised, and provides guidance for future sampling efforts to more accurately capture regional biodiversity. The framework used to build saMBA is compatible with existing global resources and is openly available, thus promoting the inclusion of other underrepresented populations to accelerate microbiome research globally.

The microorganisms living in the human gut, collectively known as the gut microbiome, have been linked to multiple aspects of human health[1,2]. Its composition is influenced by host genetics, gut physiology and by environmental factors such as diet, lifestyle, exposure to xenobiotics, among others[3–6]. Although initially the role of nature or nurture in shaping human gut microbiome composition was debated[7], it is currently accepted that while environmental factors are the main drivers, genetics makes a comparatively smaller contribution[5,8]. As geographic location represents an ensemble of genetic, environmental

and cultural factors, world-region-specific gut microbial signatures have been identified[9]. Complementary observations on migrant individuals highlight the dynamic nature of their gut microbiomes, which tend to blend with the gut microbiomes of locals in a time-dependent manner[10,11]. Additionally, the environmental forces shaping the healthy microbiome may also affect those associated with diseases, which can vary across world regions[12,13]. This regional variability challenges the universality of microbiome-derived biomarkers of disease and indicates the need for more precise microbiome-based therapeutics.

[1]APC Microbiome Ireland, Cork, Ireland. [2]Department of Anatomy and Neuroscience, University College Cork, Cork, Ireland. [3]Center for Aging Research and Healthy Longevity, Faculty of Sciences, Universidad Mayor, Santiago, Chile. [4] Center for Integrative Biology, Faculty of Sciences, Universidad Mayor, Santiago, Chile. [5]Department of Psychiatry, Amsterdam University Medical Centers Location VUmc, Amsterdam, The Netherlands. [6]Department of Psychiatry & Neurobehavioural Sciences, University College Cork, Cork, Ireland. ✉e-mail: bvalderramabobadilla@gmail.com; J.Cryan@ucc.ie

Indeed, recent efforts to define the healthy microbiome have advocated for a more even characterisation of the gut microbiomes across world regions[1].

Notably, the extent to which microbiomes from different populations are characterised is limited by their economic resources. Nowadays, more than 70% of the public human microbiome data with known origin is from people in Europe or North America, despite that their combined population represents less than 15% of the global population[14]. Consequently, entire continents are underrepresented, relying on results from research conducted in wealthier countries that may not face the same diseases that poorer countries struggle with[15], and whose main findings may fail to be generalised to other populations[13]. For instance, in South America, almost every country was identified as being underrepresented in a global-scale analysis[14]. Thus, some local[16,17] and continent-wide initiatives are trying to increase the representation of these populations[18].

In addition to increasing the sampling effort in underrepresented communities, systematically evaluating publicly available microbiome data under a unified framework is a key front to advance our knowledge about these gut microbiomes. The integrated analysis of data from different studies can facilitate the discovery of patterns that may remain elusive when studying single cohorts. In that respect, the largest compendium of human gut microbiome to date, the Human Microbiome Compendium (HMC), was created by gathering 168,000 samples from all over the world[9]. The HMC represents a big step forward in current knowledge systematisation. However, due to the scale of this analysis, the authors only included projects where more than 50 samples were sequenced[9]. Establishing such thresholds is a reasonable pragmatic decision considering the computational resources required for such analyses. However, it unintentionally impacts world regions in a way that exacerbates existing differences in sampling efforts, as poorer regions may have a higher proportion of projects with fewer than 50 sequenced samples.

To address this limitation, we screened the international nucleotide sequence database collaboration (INSDC)[19] and manually curated a list of 33 gut microbiome studies from South America, the world region with the fewest microbiome samples but one of the highest gut microbiome diversities among its inhabitants[9]. Since the workflow used to build the HMC can't be used by third parties, we recreated their workflow to analyse these previously excluded projects. Consequently, we used the same software used in the HMC, and followed the same quality criteria to filter projects, samples and sequences, thus facilitating the comparison between resources. Moreover, the workflow used to build the South American MicroBiome Archive (saMBA) is available on GitHub (https://github.com/Benjamin-Valderrama/saMBA-pipeline) and can be accessed by other researchers globally. We hope this facilitates the replication of our results and the inclusion of other neglected world regions, thus accelerating microbiome research globally.

## Results

### Introducing the South American Microbiome Archive (saMBA)
The HMC[9], a previous microbiome archive, was built by performing an automated search of gut microbiome samples from humans across different world regions. A re-analysis of the HMC resource shows clear differences in the gut microbiomes of South American populations when compared to populations from 'Central America and the Caribbean' (Supplementary Fig. 1), which suggests that explorations at a finer geographic resolution are needed. Additionally, we noted that the HMC excluded projects with fewer than 50 sequenced samples. This decision potentially impacts world regions in a way that exacerbates existing differences in sampling efforts, as poorer regions may have a higher proportion of projects with less than 50 sequenced samples (Supplementary Fig. 2). Thus, we set out to expand the catalogue of human microbiome samples from South American populations. After screening the INSDC, a manual curation of bioprojects with gut microbiome samples from South American populations (see "Methods") allowed the identification of 33 studies. The unified analysis of these studies (Supplementary Fig. 3) enabled the creation of the largest archive of gut microbiome from South American populations: saMBA. Noteworthy, ~73% of the studies were included in any archive for the first time. Interestingly, 33% of the studies included had fewer than 50 samples (Fig. 1A), suggesting that the exclusion criteria used in building the HMC increase the underrepresentation of South America, and potentially, the global south. Studies included in saMBA contain microbiome samples from 9 out of 13 South American countries (Fig. 1B, C), amounting to a total of 2913 samples in the final output (Supplementary Fig. 3), with a mean of $110 \pm 164$ samples per study, where the smallest and largest studies included 9 and 881 samples, respectively. In addition, the median number of non-chimeric reads per sample was $3.37 \times 10^4$ (Fig. 1D). Thus, saMBA represents the most comprehensive resource of gut microbiomes from South Americans.

### saMBA unveils high biodiversity across the continent
The manual curation of studies included in saMBA allowed the addition of 24 studies not included in previous archives. This highlights the need for manual curation of study metadata for more accurate assessments of regional microbiomes. By leveraging saMBA, we identify a total of 2246 genera across samples, more than doubling the current number of genera known to be present in the gut microbiomes of the region[9]. Indeed, 60% were not included as part of South American gut microbiomes in the HMC (Fig. 2A), the previously largest archive including South American microbiomes. Interestingly, 876 of the 911 genera (~96.2%) identified in the HMC were also detected in saMBA, which suggests compatibility between resources. A more extensive analysis of their compatibility is available as supplementary material (Supplementary Fig. 4). Interestingly, saMBA captures a wider range of the regional biodiversity, as shown by the density distributions of two diversity indices (Fig. 2B). This result suggests that saMBA characterises more samples at both ends of the distribution when compared to the South American samples present in the HMC.

Additionally, saMBA unveils a high biodiversity within the continent, as shown by two alpha diversity metrics (Fig. 2C, D). The median of the observed genera and Shannon indices for the continent were 72 and 2.7, respectively, which are above the global estimates represented with dashed black lines, with values of 46 and 2.1, respectively. These results suggest a high number of different genera (Observed, Fig. 2C) and a tendency towards a more even distribution of them (Shannon, Fig. 2D) in the gut microbiomes of South American individuals when compared to individuals from other world regions. Interestingly, the only region with higher gut microbiome diversity than South America is Central America and the Caribbean.

### Biodiversity across the continent is likely underestimated
We hypothesised that due to the inclusion of new samples from countries not included in previous archives, new estimations of the total biodiversity across the continent would be more accurate. A subsampling simulation analysis (see "Methods") was conducted twice: once using all samples and taxa available in saMBA, and then after filtering out rare taxa (see "Methods"). When analysing the latter, our results suggest that saMBA captures the totality of the 990 most prevalent taxa. This can be observed as the continental estimate reaching a plateau at around 2000 samples (Fig. 3A, left), which represents ~69% of the samples remaining after applying the filtering procedure (see "Methods"). As some countries have not reached the plateau despite that the continental estimate did, our results suggest that although generating more samples from those countries could reveal new taxa within those countries, those have already been observed on the continent. Note that these estimates are unlikely to be biased due to uneven sampling efforts (Supplementary Fig. 5).

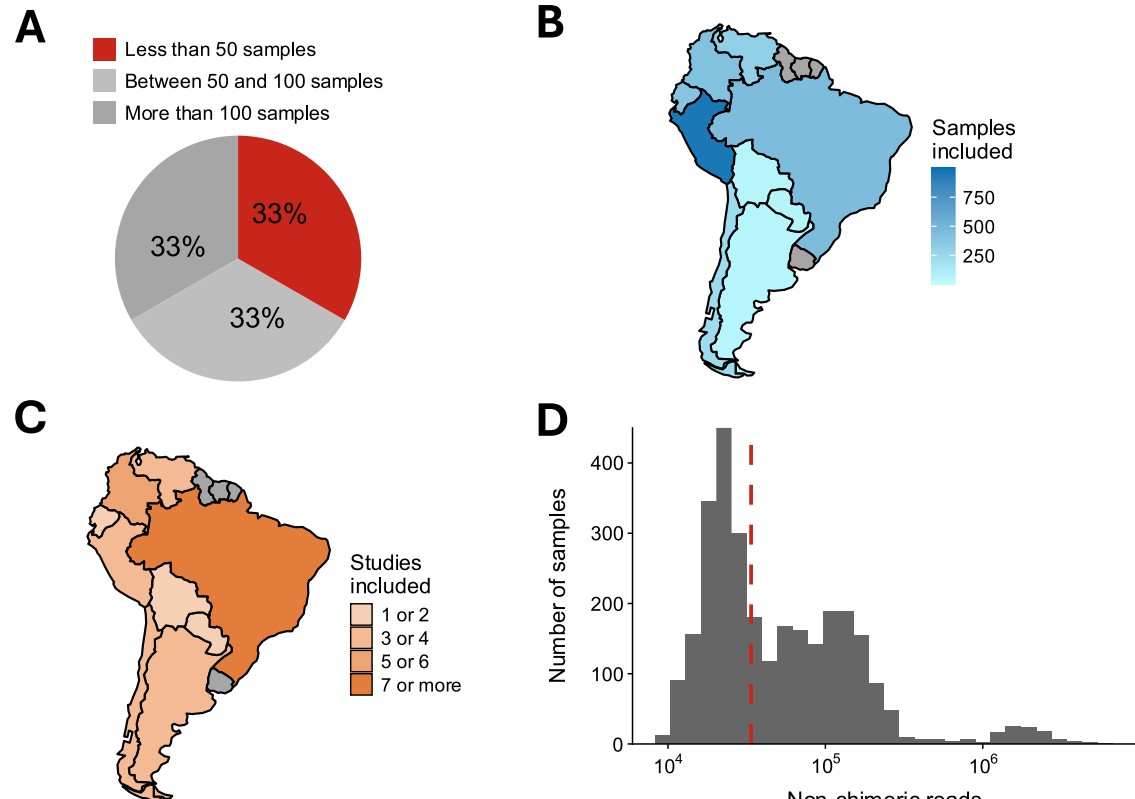

**Fig. 1 | Overview of saMBA's width across the South American continent.**
**A** Percentage of studies including less than 50 samples (red), between 50 and 100 samples (light grey), and more than 100 samples (dark grey) across countries. **B** Number of samples included in saMBA from each South American country. Darker shades of blue indicate a higher number of samples, grey indicates countries without data. **C** Number of projects included in saMBA from each South American country. Darker shades of orange indicate a higher number of projects, grey indicates countries without data. **D** Number of non-chimeric reads detected in samples across all projects included in saMBA. The red dotted line represents the median of the distribution.

Interestingly, when analysing all taxa and samples present in saMBA (without filtering), our results suggest that collecting new samples in the continent has the potential to reveal yet unobserved taxa across all countries, and thus, in the entire region (Fig. 3A, right). This can be noted as even when subsampling 3000 out of the 3110 samples included in saMBA (96%), the continental estimate of the number of observed unique taxa is still far from reaching a plateau. This is further implied by our subsampling analysis conducted on projects studying individuals living in non-industrialised settings like the Amazonian communities (Supplementary Fig. 6), showing a high yet unveiled biodiversity in their gut microbiomes. Thus, more gut microbiome samples are required to accurately characterise the regional biodiversity. However, we anticipated that sampling from different countries would have varying impacts on our understanding of South American gut microbiomes.

Hence, we aimed to identify countries that can maximise the chances of identifying yet unobserved bacterial taxa when collecting and analysing future samples. Thus, we first calculated a Local Representation Index (see "Methods"). Our results identified 6 over-represented countries where the proportion of gut microbiome samples is higher than expected based on their share of the population, and 3 underrepresented countries, characterised by the opposite (Fig. 3B). While the countries with highest Local Representation Index were Peru (4.12) and Ecuador (3.16), those with the lowest were Argentina (−9.79) and Brazil (−2.89) (Fig. 3B). Additionally, the Jaccard distance between every pair of samples was calculated for each country (Fig. 3C). Note that distances were calculated after filtering rare taxa to reduce their impact in the estimates. Higher Jaccard distances between samples indicate a lower number of shared genera.

Thus, countries with higher median Jaccard distances indicate a higher level of gut microbiome uniqueness within their population. Our results imply that future samplings should consider how well represented each country currently is, and the uniqueness of the gut microbiomes of people living in those countries. Considering both aspects to inform where to conduct newer sampling efforts can help maximise the return on investment, which is especially relevant in regions where resources are more limited.

## Discussion

Here we introduced saMBA, the largest archive of gut microbiome samples from South American countries yet created (Fig. 1), along with the code required to reproduce the analysis of this and other neglected global regions. Our results show a significant expansion in the number of genera found in samples across this world region, more than doubling what a previously released global compendium reported[9] (Fig. 2A). Additionally, by leveraging saMBA, we generated the most accurate assessment of regional biodiversity to date and provided guidelines on where new sampling efforts are most needed−an analysis that, to the best of our knowledge, is the first of its kind.

Our approach illustrates how semi-automated screening of databases (as performed in the creation of saMBA) can improve our sensitivity to include relevant datasets (Fig. 1A) not detected by fully automated approaches. This approach leads to a potentially more accurate assessment of the ranges in regional biodiversity by including more samples in both extremes of the distribution (Fig. 2B). Nevertheless, semi-automated approaches are not always feasible, as they are time-consuming, and they are more prone to error as the size of the search space increases. This further emphasises the relevance of

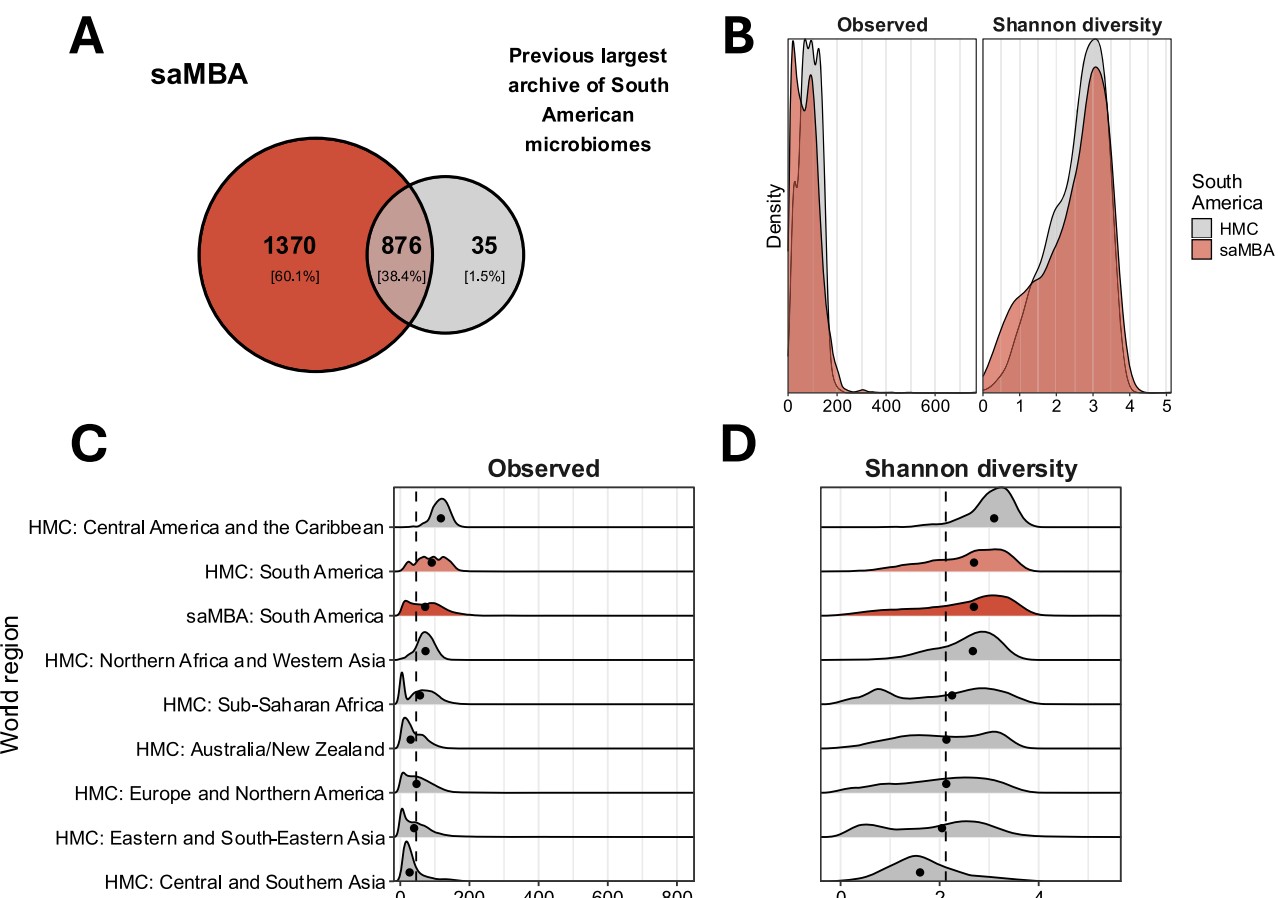

**Fig. 2 | saMBA expands the number of identified bacterial taxa and shows a high degree of diversity across the continent. A** Venn diagram showing that saMBA (in red) covers most of the already identified genera in the region by a previous global compendium (HMC)[9] (in grey) while more than doubles the current number of genera identified in the world region. **B** Density distributions of the number of observed genera (left) and Shannon diversity (right) indices for all South American samples included in the HMC (in grey) and saMBA (in red). **C** Distribution of the number of observed genera across world regions described in the HMC, and the new estimates generated in South America as compiled in saMBA. Black dots represent the median value for each region. The black dotted line shows the median value for the world. **D** Distribution of Shannon estimates of alpha diversity across world regions described in the previous global compendium (HMC), and the new estimates generated in South America as compiled in saMBA. Black dots represent the median value for each region. The black dotted line shows the median value for the world. In panels C and D, world regions other than South America are shown in grey. The distributions of the diversity indices calculated for South America are shown in pale red (when using data from the HMC), and in dark red (when using data from saMBA).

defining bounded search spaces (e.g., a single continent or even narrower categories) when creating newer compendia aiming to expand previous global efforts. Attending to this reality, the code deployed in the creation of saMBA was made public (https://github.com/Benjamin-Valderrama/saMBA-pipeline), so researchers from other world regions can generate their own archives.

By leveraging saMBA, we identified a high degree of biodiversity within the gut microbiomes across the continent, as indicated by two alpha diversity indices: Observed genera (Fig. 2C) and Shannon (Fig. 2D). Indeed, the distribution of Shannon diversities in South America follows a similar shape to that observed in a previous global compendium[9]. Our results suggest that people in this continent tend to harbour more diverse gut microbiomes than people living in most world regions. To allow for better comparisons between saMBA and the HMC, the region Latin America and the Caribbean (used in the latter) was divided into 'South America' and 'Central America and the Caribbean'. Interestingly, the only world region with higher diversity than South America is 'Central America and the Caribbean' (Fig. 2C, D). A closer inspection revealed that the only two studies from that region characterised the microbiomes from adults (PRJNA397396 and PRJNA541332), and that the sequencing depth tended to be larger than the depth observed in South American studies (Supplementary Fig. 7).

Thus, the inclusion of more studies in Central America and the Caribbean is required to better assess the real biodiversity of that region and ensure that current biodiversity is not overestimated.

Regarding the studies included in saMBA, it is interesting to note that while in some samples more than 150 different genera were identified, others showed fewer than 50. A closer inspection identified the articles from which those samples were taken. Most samples in the high end of the distribution come from one early work characterising the gut microbiome from adult individuals living in the Venezuelan Amazon[20]. Interestingly, one of the main conclusions of that work is that those gut microbiomes are much more diverse than those of people living in industrialised environments. On the other hand, most samples in the lower end of the distribution were traced back to two studies. One was conducted in infants with intestinal inflammation[21], and the other in undernourished children with diarrhoea[22].

Research conducted in individual South American countries has consistently pointed out the high diversity of the gut microbiome of their inhabitants[20]. However, accurate estimations of the true extent of this biodiversity were missing, probably due to the narrower scale of previous analyses, which focused on single populations rather than across the continent. Thus, our assessment of the regional biodiversity is the first of its kind and extent. By applying a subsampling simulation

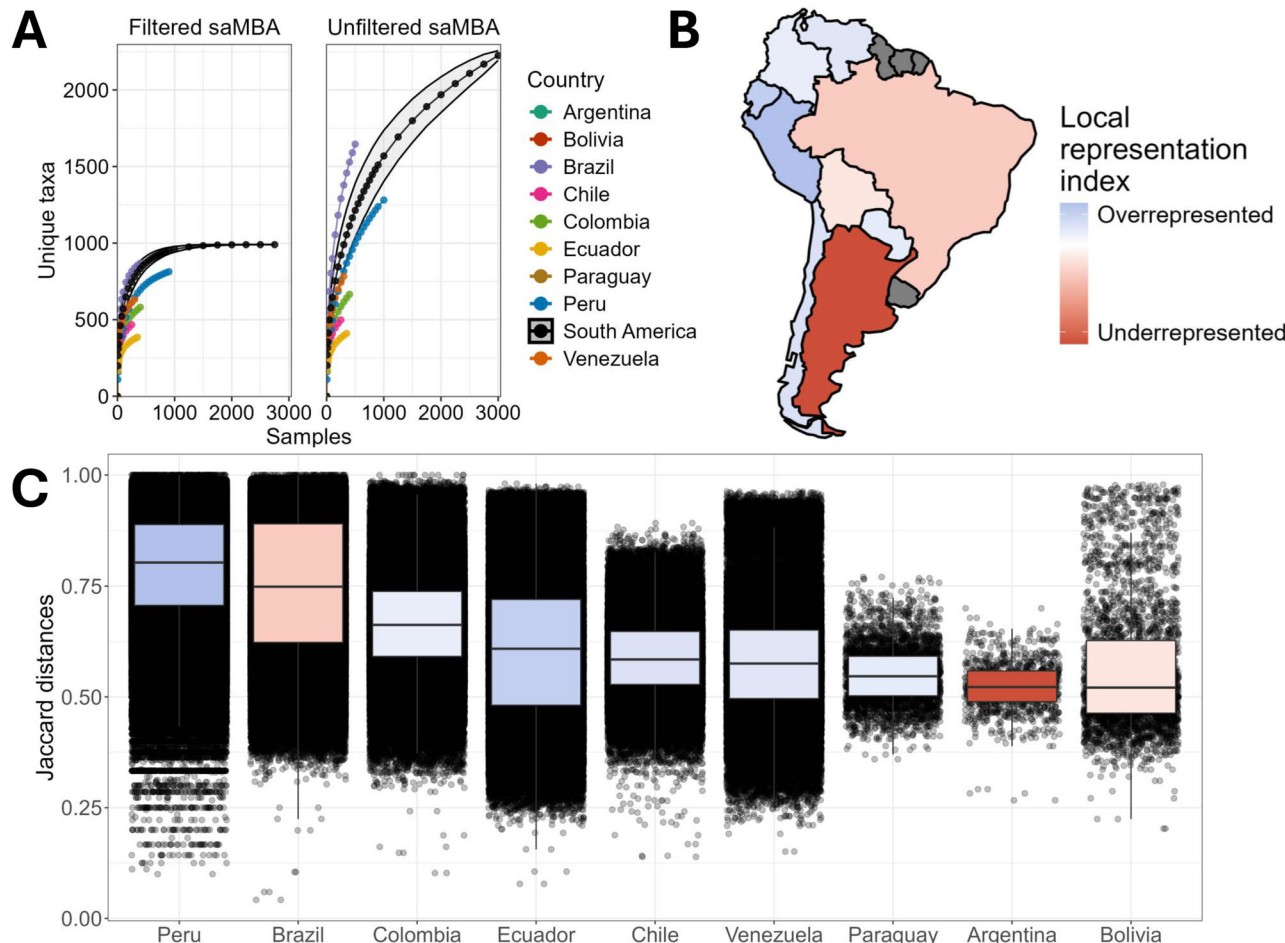

**Fig. 3 | Biodiversity across the continent is likely underestimated, and future sampling efforts should consider biodiversity and uniqueness within each country. A** Number of unique taxa identified when subsampling an increasing number of samples from each country. Each dot represents the mean value of 1000 calculations of the number of unique taxa identified when subsampling at varying depths (i.e., number of samples). Each colour is a country. The grey area around the continental estimate (black line) represents the SD. The plot on the left was built after filtering taxa and samples as described in the methods section. The plot on the right was built without any filtering. **B** Map of the Local Representation Index (LRI). Blue represents countries where the proportion of microbiome samples is bigger than their share of the continental population. Orange shows the opposite. Grey indicates countries without data. **C** Boxplot and distribution of all Jaccard distances calculated using every pair of samples available for each country after filtering taxa and samples as described in "Methods". The colours of the boxplot match the colour of the country given by its LRI.

approach (see "Methods" section), we showed that more sampling efforts are still required to accurately characterise all microbiome genera present in the gut microbiome of people living in South America (Fig. 3A). Thus, while it is well accepted that the gut microbiomes of South Americans are highly diverse, we are probably still underestimating its real biodiversity. Interestingly, previous results have shown that Latin America and the Caribbean (which encompasses South America) is the world region with the lowest number of gut microbiome samples[9]. Together, these results emphasise the need for more sampling across the continent, a challenge which new local scientific initiatives have already started to tackle[18].

However, the question of where those samples should be taken is not trivial, especially in regions where the scientific budget is scarce. Thus, we determined a Local Representation Index for each country (Fig. 3B). We aimed to assess if the proportion of microbiome samples from each country matches the proportion of the South American population living there. This allowed us to identify underrepresented countries in the region, providing essential guidance for where future sampling efforts should be conducted. Additionally, the uniqueness of gut microbiomes within each South American country (Fig. 3C) must also be considered. In this regard, we identified countries having the largest internal heterogeneity, which may indicate countries with the

highest chance of identifying yet unobserved bacterial taxa when collecting and analysing future samples.

Although these results may help in deciding where to sample next, we recognise some limitations arising from the high levels of heterogeneity among South American countries. For instance, while São Paulo city has around 44 million inhabitants, a minor yet often studied part of their population lives in the Brazilian Amazon region, with close to zero contact with the industrialised world. It's expected that these populations will differ dramatically. Additionally, social inequities– that can shape our gut microbiomes in multiple ways– are prominent within South American countries like Chile[23]. Regarding our own analysis, while we identify Argentina as being greatly underrepresented (Fig. 3B), it also shows one of the lowest microbiome uniqueness (Fig. 3C), suggesting that new samples from the country may add less novelty. However, a closer inspection reveals that the three[24-26] included studies carried out on Argentinian subjects sampled only individuals living in industrialised cities. Thus, sampling individuals living in other contexts may still bring valuable new information. In contrast, Peru was identified as overrepresented (Fig. 3B), but it also has the highest level of heterogeneity (Fig. 3C). This makes sense when considering that most samples come from individuals living in the Amazon, which suggests that further bacterial biodiversity may yet remain

to be discovered in the country. Thus, despite the progress this analysis represents, current sampling efforts conducted within each country (Fig. 3C) must be assumed to be not random, thus potentially confounding the calculated estimates of biodiversity. This further emphasises the need for finer-scale analyses accounting for diverse country-specific realities when deciding where to take future samples.

In conclusion, we built and openly share the largest archive of South American gut microbiomes: saMBA. This work contributes to a more globally representative understanding of the human gut microbiome by focusing on South America, a highly biodiverse yet often overlooked world region. The workflow deployed to build saMBA was made available to other researchers, and it is compatible with the HMC −a previously released global compendium−as they can be used together without extensive preprocessing. Thus, our work provides the impetus for the inclusion of other neglected populations to accelerate microbiome research globally and guidance for newer sampling efforts taking place in South America.

## Methods

### Identification of bioprojects included in saMBA

The European Nucleotide Archive (ENA) was used as an interface to systematically search the INSDC. Two authors (BV and PCR) screened the databases independently to identify bioprojects including samples from the gut microbiome of South American individuals. For each South American country, the following search term was used: "(microbiome OR microbiota) AND a South American country". These are two examples of terms used: "(microbiome OR microbiota) AND (Chile OR chile)" and "(microbiome OR microbiota) AND (Brazil OR brazil)". Broad search terms without restriction of publication year were used to avoid early exclusion of potentially relevant bioprojects. Each bioproject from the list of results was then screened to assess its relevance based on the project description and associated published article. Two bioprojects from Brazil (PRJEB39990 and PRJNA547608) that were not identified in the screening, but the authors knew about, were included too. All studies with samples from South American individuals that were included in the HMC were included in saMBA. Some bioprojects had microbiome samples from more than one South American country. In those cases, individual samples were allocated to specific countries using metadata from the INSDC or from the original publication. Bioprojects and samples with uncertain geographic origin were discarded. The final list of projects included in saMBA is available in Supplementary Data 1. The number of projects screened, discarded and finally included in saMBA for each South American country is available in Supplementary Data 2.

### Sample selection

Further selection of samples within bioprojects was required for reasons including: (1) some bioprojects included human samples along with environmental samples, or samples from other body sites, (2) bioprojects included samples with different sequencing strategies, like amplicon sequenced and whole genome sequenced samples, or (3) bioprojects included samples from humans living in other world regions. Thus, samples included in saMBA were limited to those flagged with the "amplicon" library strategy and that can be linked to human hosts living in any South American country at the time sampling took place, either by metadata on the original publication, or by metadata in INSDC. Samples without clear information regarding those three criteria were discarded from the analysis. The sex of microbiome sample donors couldn't be reliably identified for every sample, so the information was not included in the analysis.

Additionally, samples sequenced with technologies other than Illumina sequencings, such as MinION and 454 pyrosequencing, were further excluded. This decision was supported based on observations made by the authors of DADA2, who suggest using different parameters to analyse pyrosequenced samples[27], and because Illumina was the most common technology used among the screened bioprojects. Notably, it was observed before[9] that the sequencing instrument "454 GS" is the default value for the "instrument" field when samples analysed with Mothur[28] are uploaded to INSDC. Therefore, a manual review of the original publications associated with each bioproject was required to determine the actual sequencing technology used. If the information in the research article did not match the information in INSDC, the information provided by the authors in the article prevailed. A list of all samples used and the associated bioproject is available in Zenodo (https://zenodo.org/records/15663639).

### Analysis of individual projects included in saMBA

This workflow was built following that used in the creation of the HMC[9]. We recreated their workflow as detailed in the methods section of the article, and by implementing the steps described in the archived (and thus not reusable) code shared by the authors (https://zenodo.org/records/13733483). This includes all steps in the analysis of individual projects, and the criteria to assess the quality of the projects, samples and ASV sequences and genera included. The only deviation in those guidelines was that to build saMBA, FASTQ files from each included bioproject were downloaded using fastq-dl v2.0.4. The software, versions and code deployed to build saMBA are publicly available on GitHub (https://github.com/Benjamin-Valderrama/saMBA-pipeline). The repository also hosts instructions to deploy the workflow using a Docker image that can be accessed and downloaded from Docker Hub (https://hub.docker.com/repository/docker/bvalderrama/samba/general). Briefly, all identified projects were analysed, regardless of the number of samples sequenced. The library layout of the bioprojects (either paired- or single-end) was determined automatically by the saMBA workflow and analysed accordingly. The analysis of samples was performed in R v4.3.3 using the DADA2[27] v1.30.0 package. After the analysis, the quality of the project was assessed using the same two criteria applied by the authors of the HMC[9]: First, the number of non-chimeric reads over the total number of reads. Projects failed the first quality control step if the percentage of non-chimeric reads is <50% of input reads. A second quality control was applied. Projects failed this step if reporting 5 or more of the first 10 samples with >25% of chimeric sequences and were also discarded. Paired-end sequenced projects failing either of the two criteria were reanalysed as single-end by discarding the reverse reads, as described in the HMC[9]. The above project-level quality filters were applied due to the lack of project-level sequencing strategy information. Indeed, authors of DADA2 recommend building separate error models for each sequencing run[29], but that information was not available for all projects, and only bioproject information could be reliably inferred. Consequently, ASV inference for the entire project may not capture run-level patterns. Thus, a quality filter at the bioproject-level was used to further ensure quality in the reanalysis of each project included in saMBA. Single-end and reanalysed paired-end projects not meeting these inclusion criteria were finally discarded. Resulting ASV-level count tables of each project were clustered at the genus level, generating two main outputs for each project successfully analysed.

### Building saMBA

Genus-level count tables of each project analysed were consolidated into one archive-wide table containing 3110 samples from 33 bioprojects and 2246 taxonomic identifiers. Then, individual samples were examined and quality-filtered, as reported in the HMC[9], where taxa with a low number of reads across samples, and those present in a low number of samples were filtered out (50% and 14% removed on each step, respectively). Here, we aimed to remove similar proportions of taxa on each step. First, 139 samples with fewer than 10,000 non-chimeric reads were discarded. Then, 1123 taxonomic entries (50%) were removed as they had fewer than 80 reads across samples. We then removed the other 133 taxa (12%), as they were present in less than

3 samples. After removing rare taxa, we aimed to remove samples with fewer than 10,000 remaining reads, but none were discarded. Finally, 58 samples with more than 10% of reads unclassified at the phylum level were also discarded. Therefore, the filtered count table contains 2913 samples and 990 taxonomic identifiers. The consolidated genus-level count tables before and after filtering samples and taxa, as well as the consolidated ASV-level count table without preprocessing, can be accessed online through Zenodo (https://zenodo.org/records/15663639). The GitHub repository with the workflow contains the code used to consolidate the genus-level count tables of each project into one archive-wide table, and the code used to generate the genus-level filtered count table.

### Analysing saMBA

The consolidated and filtered genus-level table was then used in all analyses except those performed to compare saMBA to the HMC. Analyses were performed in R v4.3.3. To estimate microbiome diversity, the Observed genera and Shannon alpha diversity indices were calculated using the package microbiome v1.24.0. To estimate the per-country microbiome uniqueness, distances between pairs of samples within each country were calculated using the Jaccard dissimilarity as implemented in the function vegdist from the package vegan v2.6.6.1. Note that distances were calculated after filtering taxa and samples, as described in the methods, to limit the effect of extremely rare taxa in the estimation of pairwise distances. Figures depicting geographic data were generated using the package spData v2.3.3. Other figures were generated with ggplot2 v3.5.1 and patchwork v1.3.0. The scripts deployed to analyse saMBA and to generate the figures of this article are publicly available in an independent GitHub repository (https://github.com/Benjamin-Valderrama/saMBA-article).

### Local Representation Index

The calculation of the index was performed as described before[14]. Briefly, global population data was obtained from the World Bank report for the year 2022, as it contains the most updated and trustworthy information we could find. That information was then used to calculate the country's share of the total South American population. Another percentage was calculated for each country, indicating the country's share of the total amount of samples included in saMBA. Then, we calculated the Local Representation Index (LRI): for countries with a share of samples higher than their population share, we divided the former by the latter. In the opposite case, the LRI was calculated as the negative reciprocal of this number. Thus, positive numbers represent how many times the number of samples is higher than what is expected based on the country's population share (i.e., the country is overrepresented in saMBA). On the other hand, negative numbers represent the factor by which the number of samples from that country needs to be increased to match the country's share of the population (i.e, the country is underrepresented in saMBA). Notice that values calculated as described above were used to generate the relevant plots, which is different from the previously published approach[14], where the log10 of the values was used for visualisation.

### Subsampling analysis to estimate regional richness

To estimate how the current sampling effort allows for uncovering regional microbiome richness, a custom subsampling simulation approach was used. First, counts were transformed into binary data, where 1 represents bacterial taxa present in the sample, and 0 was used otherwise. Then, a custom function was used to perform a consecutive subsampling by randomly selecting an increasing number of samples and estimating the mean number of novel taxa identified across all selected samples over 1000 iterations (specific sampling depths are described on https://github.com/Benjamin-Valderrama/saMBA-article). This process was conducted for each country independently, and then for the entire continent.

Importantly, the number of samples used in each iteration was below the total number of samples available for that country or the continent. This subsampling process was applied to the saMBA dataset both before and after filtering samples and rare taxa to generate Fig. 3A (also shown as Supplementary Fig. 5A, D). The same process was then repeated on the same two datasets with the addition of an intermediate rarefaction step. Two rarefaction regimes were used to evaluate the effects of uneven sampling depths across countries on regional estimates of biodiversity: in one iteration, samples were rarefied to 9000 reads (Supplementary Fig. 5B, D) and in other, to 1000 reads (Supplementary Fig. 5C, E) before estimating the mean number of unique taxa across the 1000 iterations used for each subsampling depth.

### Subsampling analysis estimates non-industrialised microbial richness

When available, the manuscripts associated with the projects included in saMBA were screened to identify if the subjects from whom samples were taken lived in industrialised or non-industrialised settings. If the manuscript wasn't available, information deposited on the INSDC was used. When a clear identification of the setting couldn't be made, the project was not included in this analysis. Projects analysing samples from non-industrialised settings were further divided into 2 groups depending on whether the research topic included an examination of individuals suffering from a disease or diarrhoea, and studies characterising microbiomes of individuals without those conditions. Note that the categorisation was done at the level of projects, not samples, as it was not possible to achieve so for all samples due to limited metadata information. For each group, we performed a subsampling approach as described in the section above to estimate the richness revealed by current sampling efforts. The goal was to characterise the biodiversity of gut microbiomes in non-industrialised settings, which are generally deemed as more diverse. Additionally, it is recognised that people from non-industrialised settings are more exposed to pathogens and parasites, which affect the composition of their gut microbiome[30], hence the subdivision between studies conducted on individuals with and without the conditions mentioned above.

### Comparing saMBA and the human microbiome compendium

It has been reported that using different workflows to analyse the same biological sample can modify the interpretations of the data[31]. To validate compatibility between saMBA and the Human Microbiome Compendium (HMC), we evaluated the similarity of the results given by the two workflows when analysing the same samples. Note that the HMC count table that users can access corresponds to the dataset before applying the preprocessing steps described in the 'Building saMBA' section. Therefore, to ensure fair comparisons, the unprocessed saMBA count table at the genus level was used when comparing saMBA and HMC (Fig. 2A–C and Supplementary Fig. 4). All other results were generated using the filtered (processed) saMBA count table. Using R v4.3.3, the 7 most abundant phyla across all samples from South American subjects included in the HMC v1.0.1 were identified, and their relative abundances were determined for each sample analysed on both workflows. This allowed a qualitative assessment on the degree of similarity of the same sample analysed with the two workflows. Then, a quantitative assessment was sought. This time, Bray-Curtis dissimilarities were calculated using genus-level count tables and the R function vegdist from the package vegan v2.6.6.1 for each pair of samples analysed with both workflows. Dimensionality reduction was performed using the R function prcomp. Additionally, a Mantel test was performed using the function 'mantel' from the vegan package and the calculated Bray-Curtis dissimilarity matrices as inputs. This test was used to determine the correlation of the HMC and saMBA matrices using the taxa and samples shared between resources.

## Reporting summary

Further information on research design is available in the Nature Portfolio Reporting Summary linked to this article.

## Data availability

Raw sequencing data used in this article is publicly available and can be accessed through the European Nucleotide Archive (ENA). The accession codes of each reanalysed sample can be found in Zenodo: https://zenodo.org/records/15663639.

## Code availability

The code used in the workflow deployed to download and analyse each microbiome study can be found in this GitHub repository: https://github.com/Benjamin-Valderrama/saMBA-pipeline. The code used to generate the analysis and figures included in this article can be found in this GitHub repository: https://github.com/Benjamin-Valderrama/saMBA-article.

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

## Acknowledgements

APC Microbiome Ireland is funded by Research Ireland (previously Science Foundation Ireland, SFI grant code 12/RC/2273_P2). BV would like to thank Dr. Richard J. Abdill for the helpful discussions by e-mail about the implementation of the workflow used to build the Human Microbiome Compendium. BV thanks Dr. Samuel Miravet-Verde for the discussions about the automated screening of the INSDC.

## Author contributions

B.V. conceptualised the project together with P.C.-R., T.F.S.B., A.L., G.C. and J.F.C. B.V. and P.C.-R. conducted the screening and identification of relevant bioprojects. B.V. curated the final list of bioprojects included. B.V. wrote the code to build saMBA. B.V. performed the data analysis. B.V. prepared the first draft. B.V., P.C.-R., T.F.S.B., A.L., G.C. and J.F.C. read and edited the manuscript. G.C. and J.F.C. acquired the funding.

## Competing interests

J.F.C. has been an invited speaker at conferences organised by Bromotech, Yakult and Nestle and has received research funding from Nutricia, DuPont/IFF, and Nestle. G.C. has received honoraria from Janssen, Probi, Apsen, and Ingelheim Boehringer as an invited speaker; is in receipt of research funding from Pharmavite, Fonterra, Reckitt, Nestle and Tate and Lyle; and has been paid for consultancy work by Yakult, Zentiva, Bayer Healthcare and Heel Pharmaceuticals. This support neither influenced nor constrained the contents of this manuscript. The remaining authors declare no competing interests.
