## [Transparent Peer Review file · Nature Communications]

The South American MicroBiome Archive (saMBA): Enriching the Microbiome Field by Studying Neglected Populations

Corresponding Author: Mr Benjamin Valderrama

Version 0:

Reviewer comments:

Reviewer #1

(Remarks to the Author)

Valderrama et al. describe here a new integration of 33 microbiome projects from South and Central America. They describe the content of the samples and patterns at the country and continent levels, and discuss the implications for underrepresentation of populations from these regions relative to the rest of the world. Their approach is reasonable, consistent, and clearly described. Their analysis code is publicly available, and their pipeline is well-documented. They describe several interesting conclusions, and their model provides an example (and plenty of computational resources) for others to do similar work focused on other populations. I believe this is useful work that merits publication, pending necessary revisions to the text and possibly analysis.

Their results are mostly descriptive, which is not necessarily a negative factor. However, much of their justification for the work, as stated in the introduction and discussion, is in how the dataset compares to the Human Microbiome Compendium (HMC). The paper perceptively notes that the compendium's sample-filtering process excluded many samples from the region, but this manuscript would be stronger with a more quantitative consideration of these differences and a consideration of the implications.

For example, the paper notes that the HMC's exclusion of projects with fewer than 50 samples "unintentionally impacts world regions in a way that exacerbates existing differences in sampling efforts, as poorer regions may have higher proportion [sic] of projects with less than 50 sequenced samples." This would be a compelling finding, but it appears to be a guess, and according to Figure 1B, most of the projects included in saMBA also include more than 50 samples. It's not critical that this specifically be investigated, but it is one example of the manuscript raising questions that it does not address in its analyses.

MAJOR NOTES:

Line 23–25: The paper states that this dataset is meant to address a sampling disparity that currently "limits generalizability and reduces the performance of predictive models." This may be intended as background information, but if a problem like that is stated in the abstract, it suggests this dataset will improve the performance of predictive models. This is not addressed in the results. In short, what are the implications of this data being available now? What analyses does it improve, and what could it be used for? Is there any analysis that would demonstrate this?

Lines 153–162: This is a clear and useful description of the paper's filtering protocol, but it would be very helpful if it also described the effects of these steps, particularly because these steps are compared so directly to filtering from the HMC paper. The paper notes that the filtering has a more stringent effect on the smaller saMBA dataset, but I'm concerned it's having a much more dramatic effect than it might otherwise appear. For example, when filtering out taxa with fewer than 1000 reads across all samples, the HMC paper states that 50 percent of taxa were removed. The HMC dataset has about 57 times as many samples as saMBA and a higher median reads per sample, so I would guess the proportion of removed taxa would be far higher—1000 reads seems like it would be easier to find in the HMC's (very roughly) 6.9 billion reads compared to saMBA's 0.4 billion reads. So if the HMC analysis was based on the top 50 percent most prevalent taxa, what would that number be for saMBA? Even when evaluating a single dataset, I would guess results from the 5 percent most prevalent taxa would look different than the results from the 50 percent most prevalent. I may be thinking about this the wrong way, but if the goal is to match the filtering done in the two datasets, I think it would be more appropriate to set the cutoffs to a level that

achieves a similar proportion, rather than using the same cutoff.

Lines 197–208: I may be misinterpreting the method here, but I believe the approach as described does not account for sampling effort, which could be artificially inflating the richness of individual countries. For example, we see that at 100 samples, the number of novel taxa observed in Ecuador is (on average) much lower than the number observed in Peru. The number of samples from each country is identical in this comparison, but if the projects in Peru sequenced their samples with, say, five times as many reads, then it may be misleading to say that the microbiomes of one region harbor more novel taxa than the other—one of those countries was just captured at much higher resolution. I believe one way to mitigate this would be at least similar to the approach taken in the referenced HMC paper—samples were randomly rarefied to 1000 reads each prior to the sampling process described here. A chart using this data may look very different, but it would be comparing countries on more even footing: If you took, 1000 reads from Peru and 1000 from Ecuador, which would contain more novel taxa? It doesn't look like the HMC analysis randomized the samples for every iteration, but the main point is simply that it could be much more interpretable to account for number of reads per sample here.

I realize that could be a computationally intensive effort; one alternative, but not the only one, could be to just interpret this plot slightly differently by saying it assumes disparities in reads per sample will continue as they currently are.

Lines 260–262: The statement here reads, "the exclusion criteria used in building the HMC increases the underrepresentation of South America, and potentially the global south." This may be true, but it appears to be unsupported by the results, at least as presented. Though "30% of the studies included had less than 50 samples," this says nothing about the proportion relative to other regions. It seems to me that "underrepresentation" would only happen if other world regions have a differing distribution of samples per study: If 30% of South American studies have less than 50 samples, and, say, 60% of studies from Europe, a 50-sample cutoff would inflate South America's representation, not the other way around. This is a compelling question, but if it's included here, it should be better supported.

Lines 291–294: The use of alpha diversity measures here, particularly Chao1, seems at odds with the paper's description of the filtering of low-abundance and low-prevalence taxa. It seems filtering out those taxa would disrupt the relationship between the two measures presented. In addition, there have been concerns raised (DOI 10.1093/ismejo/wrae106) about the Chao1 estimator being used not just for filtered data such as this, but for some amplicon analyses in general. In short, if Chao1 is attempting to estimate unseen taxa, but many have already been filtered out, it's unclear what this might indicate; the paper would benefit at least from some elaboration here.

Lines 311–312: I am a little confused by the logic in the paper's interpretation of Figure 3A, which states that "although some countries are close to reach [sic] a plateau, none has done so yet, suggesting that newer samples taken in most South American country [sic] will likely identify new genera not yet included in saMBA or any other compendium." It's unclear to me how this conclusion is reached, particularly because the "continental estimate" in Figure 3A has plateaued. If the lines for each country continue to rise, but the pooled results ("South-america" in 3A) are no longer growing, this could also suggest that most South American taxa have been observed (hence the flat black line), but that those taxa are present at different prevalences in each country. If this were the case, then it may be that further sampling from Bolivia would reveal a taxa novel to Bolivia, but that would already have been observed in multiple other countries. Even if the continental estimate had not leveled off, however, it's not clear here why a taxon novel to South America would "likely" be unobserved "in any other compendia." It seems possible, but if there are unidentified taxa in a region, how would one discern the probability that those taxa have not already been observed elsewhere?

Line 463: This section states that "world regions used in previous work are too broad," but it is unclear what this means—"too broad" for what? The fascinating result from Supplementary Figure 1 supports the statement that "explorations at a finer geographic resolution are justified," but if the criteria for justifying these explorations is that between-group differences can be identified, this chain could continue down the line: maybe continental regions are too broad because they fail to account for urban vs rural differences, and those classifications could be too broad because of differences between landlocked cities and those on the coast. If the regional level is "too broad" but the level presented here is correct, the manuscript would benefit from an explanation of why this specifically is the correct level.

MINOR NOTES:

Line 140: This notes that the "v1.30.0" version of DADA2 was used here, but it appears the most recent version of that software is 1.26.

Line 224: It would be helpful if the statement about pathogen/parasite exposure's effect on the microbiome was supported by a reference.

Lines 155–157: I was confused by the statement here that "the HMC remove taxa present in less than 1,000 samples," because it doesn't seem to align with the filters in the previous sentence (taxa with fewer than 1,000 reads across samples, or taxa present in less than 100 samples). It also doesn't appear to describe the HMC filtering referenced: "To reduce sparsity introduced by exceedingly rare taxa, we then removed 2018 taxonomic entries (50%) with fewer than 1000 total reads across all remaining samples, and a further 578 taxa (14% of the original total) that were detected in fewer than 100 samples." The manuscript would benefit from clarification here.

Lines 185–195: The Local Representation Index is a useful tool here, but the description as written is difficult to follow. Phrases like "sample percentage" and "how many times the number of samples needs to be increased" could benefit from rephrasing.

Line 263: This states that 9 of 13 countries were present in the dataset—does this refer to the filtered or unfiltered data? If the former, it would be helpful to know if other countries were present but got filtered out.

Line 264: I believe this is the only point in the text where the number of samples is specified. This is an important enough number to put in the abstract, not just on the bottom of page 7.

Supplementary Figure 3C: If this plot was made using a dissimilarity matrix, would it be PCA (principal component analysis), as stated, or PCoA (principal coordinates analysis)?

(Remarks on code availability)

The analysis scripts for the paper are straightforward and well-commented. I wasn't able to find the code used for filtering samples and taxa, but this is likely user error on my part. The repositories, and associated wikis and website, are exemplary.

Reviewer #2

(Remarks to the Author)

The manuscript by Valderrama et al. aims to generate a resource focused on highlighting studies related to the South American Human Gut microbiome. This resource is intended to facilitate comparative analysis while implementing open-access, reproducible bioinformatic pipelines.

The work is of significant interest, as it has been reported that Latin American data are underrepresented in human gut microbiome studies. The manuscript adequately emphasizes that the most recent reports, while applying filters based on the number of samples per study to improve computational efficiency, simultaneously introduce bias against studies with smaller sample sizes—an issue particularly relevant to regions with limited resources.

My main concern relates to the search, selection, and filtration of studies and samples used in the final analysis: Although I agree that this is the largest compendium of microbiome studies from South America to date, it does not constitute a comprehensive collection of all studies that have published South American microbiome data, and this limitation should be explicitly acknowledged. Furthermore, while the effort to ensure transparency and reproducibility in all steps is commendable, there are still improvements that could be made in this regard.

For example, it would be valuable to list the raw results from the initial search and to show how successive filtration steps reduced the number of studies, leading to the final list presented in Supplementary Table 1. A representation similar to Supplementary Figure 2 could be expanded to include the list of studies and all filtering steps, both prior (e.g., sequence type) and subsequent (e.g., abundance, prevalence).

In line 78, the manuscript states that the authors “screened the INSDC” and manually curated a list of 33 gut microbiome studies. However, Supplementary Table 1 shows that some studies were retrieved from ENA, while others came from Google Scholar or the Human Microbiome Compendium.

The list in Supplementary Table 1 includes studies using both amplicon and shotgun datasets. Furthermore, on the amplicon studies, although most projects used Illumina sequencing, some used 454. However, the Methods section under “Sample selection” states that such datasets were removed. Clarification is needed regarding the final list of studies and samples included in the analysis.

An additional filtration step states that “samples with more than 10% of reads assigned to Archaeal taxa or with more than 10% of reads unclassified at the phylum level were also discarded.” The rationale behind this filtering criterion should be clearly explained.

Similarly, the manuscript states: “Second, projects with 5 or more of the first 10 samples with >25% of chimeric sequences were also discarded.” It would be helpful to justify why a project-level filter was applied here rather than a sample-wise filtering approach.

Here is your corrected text:

Other concerns:

1. The manuscript does not address potential batch effects in the analysis. The Human Microbiome Compendium (HMC) claims that batch effect correction is unnecessary due to the large number of studies analyzed, which allows biological variation to dominate. However, in this work, the number of studies is more limited. Therefore, the potential need for batch or study effect normalization should be discussed.
2. The selection of beta-diversity metrics requires clearer justification. While Jaccard dissimilarity is used in one section, Bray–Curtis dissimilarity is employed for comparison with HMC results. High within-country Jaccard distances are interpreted as indicative of microbiome uniqueness; however, this metric is sensitive to rare taxa, which may be prevalent in several of the included studies. These values can be influenced by sequencing depth, compositional sparsity, and inter-

study technical variability. These factors should be considered in the discussion.

3. The use of the Local Representation Index (LRI) is interesting, but it may assume that sampling is random across the population, which is not the case for any of the included studies.

4. In the comparison between different workflows, there are established metrics, such as Procrustes analysis or mantel tests, to assess whether beta-diversity profiles are equivalent. It would be informative to investigate the samples with Bray–Curtis dissimilarity greater than 0.25 between workflows. Was there any pattern associated with these discrepancies?

5. In the analysis of non-industrialized settings, although the number of samples is reported (1,072 vs. 446), the number of studies or sampling sites is not provided. This information could significantly affect the interpretation of the results and should be clarified.

6. The terms “metagenome,” “metabarcoding,” and “shotgun metagenomics” are used interchangeably throughout the manuscript, which may lead to confusion. Defining these terms early in the text would improve clarity.

7. The classification of “industrialization” requires further refinement. South America includes both highly industrialized urban centers and remote rural regions. However, the manuscript appears to generalize industrialization status at the country level. While classifying samples by industrialization can offer ecological insight, metadata quality and consistency vary widely across projects. In some cases, assumptions about rural versus urban settings may oversimplify complex lifestyle gradients.

Minor comments:

1. Why is Figure 1B cited before Figure 1A?

2. Line 264: Given that the standard deviation is larger than the mean for the number of samples per study, it would be helpful to show the full range or distribution.

3. Line 290 states that the analysis “unveils a high biodiversity,” but without comparisons or references, it is difficult to assess whether the reported values are indeed high.

4. Line 311: “South American country” should be “South American countries.”

5. Figure 3A: The colors used for Bolivia and Venezuela are difficult to distinguish. Consider improving the color contrast.

6. Lines 94–97: The search strategy is clearly described, but it lacks reproducibility. Consider including the exact search strings used for each country and the date(s) of access to the ENA database.

7. Lines 106–115: It is unclear whether samples with ambiguous metadata (e.g., uncertain geographic origin) were included or excluded. Please clarify the exclusion criteria related to metadata uncertainty.

8. Lines 219–225: Clarify how disease and non-disease samples were separated. Was this distinction based on project metadata or manually extracted from publication abstracts? Also, indicate whether any filtering was performed for co-morbidities.

(Remarks on code availability)

Comments on the saMBA pipeline (<https://github.com/Benjamin-Valderrama/saMBA-pipeline/wiki>):

1. The command to create the environment uses an outdated file name:

```
micromamba env create --name samba --file env/samba.yaml
```

This should now reference `samba.yml`.

2. The current environment specification may lead to dependency resolution issues:

Could not solve for environment specs

The following packages are incompatible:

```
└─ _libgcc_mutex ==0.1 conda_forge does not exist (perhaps a typo or a missing channel);
```

```
└─ _openmp_mutex ==4.5 2_gnu does not exist (perhaps a typo or a missing channel);
```

```
└─ binutils_impl_linux-64 ==2.40 ha1999f0_7 does not exist (perhaps a typo or a missing channel);
```

This suggests that the `samba.yaml` file in the repository should be updated. It would help to refresh the environment file using more stable channel instructions (e.g., include `noarch`, review Bioconda priorities, and ensure compatibility with `conda-forge`). It should also be noted that the latest `mamba` solver (v2.0.0) introduced significant changes, which have caused many Bioconda packages to fail. A warning or guidance for users on this point would be useful.

3. The current environment file includes 324 dependencies. Reducing the number of packages, where possible, would improve reproducibility and reduce potential for conflicts.

4. The instructions mention that installation and environment setup may vary. This section should point to troubleshooting resources or include common alternatives to improve user support.

5. Consider making the reproducibility-related instructions more prominent, such as placing them in the main README. The Wiki tab is often overlooked in GitHub repositories.

Reviewer #3

(Remarks to the Author)

The overall focus of the work is to amend/expand the HMC compendium with a manually curated selection of south American microbiomes from the public repositories. This is of interest to the community as the region remained under-sampled in that compendium despite its previously observed high biodiversity. The work also provides valuable results that could guide future human gut microbiome sampling campaigns in south America, along with publicly available bioinformatic pipelines that can be applied to other continents in a similar fashion.

On the other hand, the manuscript’s claims that it “deepens our understanding of the different stable states of the human gut microbiome”, “generated the most accurate assessment of regional biodiversity to date” or expanded “the concept of the healthy microbiome to be more globally representative” are not sufficiently substantiated by the results; the proposed expanded compendium could be used to that end indeed, but the analyses and scientific results to substantiate such

sentences are not sufficiently present in the manuscript itself.

The presented analyses are sound, along with its interpretations. The bioinformatic methodology is adequate and mostly well described.

The code and datasets included in the repositories are well presented and clear. In fact, the authors' efforts towards transparency and open science are commendable, including the public availability of the pipeline, the extensive documentation, and the invitation to external researchers to directly share new studies for their inclusion in the database. Nonetheless, we spent a considerable amount of time trying to install the saMBA pipeline on Linux, but ultimately we were unsuccessful. We're still unsure whether the issues stemmed from some incompatibility on our end, but we tried several troubleshooting steps: installing micromamba instead of using conda, removing conda's activation steps from the `~/.bashrc` file altogether, resetting the PYTHONPATH to an empty string, removing some possibly conflicting packages from our system, and even simplifying the `.yml` file by removing version constraints and all R packages. None of these attempts resolved the problem. Perhaps the pipeline runs smoothly on a freshly installed Linux system, but in our case we were unable to get it working. We also tried running the script without installing anything, but without `fastq-dl`, which we also failed to install, we weren't even able to download the test files. As other researchers could face similar issues, a possible option would be to provide a Docker container, which is fully self-contained and would ensure reproducibility and usability.

The author's should check whether their research follows the journal's 'Sex and Gender Equity in Research – SAGER – guidelines'. For instance, i) they should mention that the sex and/or gender of the metadata likely arises from self-report, and ii) "Data should be reported disaggregated for sex and gender where this information has been collected".

SPECIFIC COMMENTS:

Mayor comments:

L28. It would seem that the results provided on this subject are not related to other populations. Thus, what does the term "high" actually mean here if it is not properly compared to other studies? Are these results normal if compared to other global populations; are south-american microbiomes particularly rich and unique?

L29. "expanding the concept of the healthy microbiome to be more globally representative" I do not see this sentence as related to the work. Indeed, the curated database could be used to that end, but there are no particular related results on the matter within the manuscript.

L33. "The framework used to build saMBA is compatible with existing global resources". Further information on comparability (see comments below) and compatibility is required. For the latter, what do the authors mean by "compatible", is it that one can download profiles from both resources and analyzed them together without extensive processing?, how?.

L81. Can you expand on the issue? What is available at HMC (e.g. filtered genus-level count tables, unfiltered ASV tables, fastq sequence and metadata)? What is not available with HMC?

L151-156. Where the genus-level tables, from which most analyses derive, normalized or subsampled in any way? (e.g. by subsampling to a fixed sequencing effort). Most analyses may be biased otherwise (e.g. Bolivian samples in fig3A my harbor more taxa than those of Ecuador due bias in the sampling depths of both projects, or the use Jaccard distances to estimate microbiome uniqueness within countries; it is unclear how "presence" was defined across samples with differing sequencing depths.)

L235. Why were these analyses carried out at the phylum level and not at the genus level (or both)? If available HMC data is available only at the phylum level place state so. However, it would seem from L287 that such information is available. If the phylum level was chosen so that Suppl. Fig. 1 provides readable results OK, but the analyses in suppl. Fig. 3 (BC) should be carried out at the genus level, as phylum level is extremely coarse-grained.

L290-303. This seems to be too descriptive in nature. The information contained in panel A conveys the authors' idea that saMBA expands previous knowledge on south American microbiomes. However, the information depicted in panels B and C lacks sufficient context. How do those medians and distributions compare to microbiomes from other parts of the world (processed and analyzed in the same way)?

L359. "Additionally, by leveraging saMBA, we generated the most accurate assessment of regional biodiversity to date". The curated database could be used to that end, but there are scarce results on the matter within the manuscript.

L377. And also continuing my previous comment on the matter; a joint analysis of these results against other world regions (processed and analyzed in the same fashion) would be valuable. Similarly, (L378) the authors could have derived Chao1 and Shannon values from the previous compendium to compare using the available count table mentioned in L230.

L385-387. So saMBA contains samples regardless of the health status? How does this impact the results? How many were from "unhealthy" individuals? How does this relate to the values reported for genera and diversity indices?

L452. "saMBA' deepens our understanding of the different stable states of the human gut microbiome". I do not see this statement as related to the content of the manuscript. I agree that saMBA could be used to that end, but the results presented do not particularly relate to that sentence, but are rather directed to exemplify how saMBA outperforms HMC in terms of south American samples.

L455. What do you actually mean when you say that the workflow is compatible with the HMC? Please describe succinctly.

Supplementary figure 3C should link (with ellipses, or better with a line) each pair composed of profiles from the same sample analyzed with both pipelines (for genus level analysis). Additionally, I would suggest performing some kind of permutation test or cumulative distribution to better gauge the between-sample dissimilarity (technical) vs among-samples dissimilarities (biological); supplementary figure 3B is too broad scale. E.g. for how many samples does the least dissimilar profile not belong to the same sample but with different processing pipeline?. Or if using per sample empirical cumulative distributions: what is the average and SD of the position that the technical (bioinformatic) replicates occupy along the complete dataset?. This should be done at the ASV or genus level.

Minor comments:

L42-44. What about host physiology, does it have a role in shaping the gut microbiome?

L49-50. While I follow and agree on the rationale of the sentence, I do not see that the previous observations “indicate that healthy microbiomes and those associated to diseases are different among world regions”. Consider rephrasing.

L140-146. I could not find in the original HMC publication indication that “Projects were discarded if the percentage of non-chimeric sequences is <50% of input sequences” but maybe I am mistaken. On the other hand, what is the rationale for this procedure? I do not fully understand not only the rationale but how it interacts and affects the next filter. Are we talking about ASVs or sequence counts? Are those values related to the total of ASVs sequences or to the total number of sequences belonging to chimeric ASVs. Please clarify in the text.

L154. Just to clarify, by “taxa” you are referring to genera right (not ASVs)?, if so, consider if it is better changing to “genera”.

L154-157: I encourage the authors to include a short discussion on the possible impact of this more stringent threshold. If the applied threshold were to be proportional to the 1000-samples one applied in the case of the HMC, then it should be about 17 or 18 samples. However, a threshold of 100 samples was chosen, making it more than 5 times more stringent. Furthermore, if I understand correctly, the filtering approach appears to exclude approximately 3.8% of the known genera that were included in the HMC originally according to lines 286-287 — a non-negligible loss of richness, given that 24 additional studies are being included in saMBA.

L162- It would seem from the previous sentences that all the filtering was done with the genus-level table; is that so? Is the ASV count table mentioned here the same as was mentioned in L148? Is this ASV count table unfiltered or has it been filtered with the same pipeline described in L153-160? It would seem from L163-164 (and the repository mentioned) that this is the case, so this is just a minor clarification I need.

L230. “Note that the available HMC count table is from before the preprocessing steps described in the ‘Building saMBA’ section. Therefore, to ensure fair comparisons, the unprocessed saMBA count table at the genus level was used when comparing saMBA and HMC (Figure 2A and 233 Supplementary Figure 3).” Why wasn’t the HMC genus level count table filtered as described in the ‘Building saMBA’ section before the comparative analyses instead? I understand the sentence and those that follow, but feel that I may be missing something with regards to the nature of the HMC count table (see also comment indicating that more information on the nature of HMC data is called for).

L322-323. What do you mean? In what sense? Consider expanding the argument just a little bit (or remove the sentence).

L366-368. Why restrict the space to continents? The argument feels unsupported and related to what the authors present and not a particular scientific question of interest. You could argue as the authors did early in the manuscript that “As geographic location represents an ensemble of genetic, environmental and cultural factors,” (if so consider reformulating the statement on these grounds). However, wouldn’t it be better to limit the search space to meaningful populations with different genetic, environmental and cultural factors? The authors actually delve on this on L414-L432.

L387-389. This is expected, I would not consider the use of the term “validation” in this scenario.

In Figure 3, I suggest including the actual numerical range of the represented LRI values, at least in the caption or the main text. This would help the reader interpret the scale and relevance of the observed differences.

(Remarks on code availability)

The code and datasets included in the repositories are well presented and clear. In fact, the authors’ efforts towards transparency and open science are commendable, including the public availability of the pipeline, the extensive documentation, and the invitation to external researchers to directly share new studies for their inclusion in the database. Nonetheless, we spent a considerable amount of time trying to install the saMBA pipeline on Linux, but ultimately we were unsuccessful. We’re still unsure whether the issues stemmed from some incompatibility on our end, but we tried several troubleshooting steps: installing micromamba instead of using conda, removing conda’s activation steps from the `~/.bashrc` file altogether, resetting the PYTHONPATH to an empty string, removing some possibly conflicting packages from our system, and even simplifying the `.yml` file by removing version constraints and all R packages. None of these attempts resolved the problem. Perhaps the pipeline runs smoothly on a freshly installed Linux system, but in our case we were unable to get it working. We also tried running the script without installing anything, but without `fastq-dl`, which we also failed

to install, we weren't even able to download the test files. As other researchers could face similar issues, a possible option would be to provide a Docker container, which is fully self-contained and would ensure reproducibility and usability.

Reviewer #4

(Remarks to the Author)

(Remarks on code availability)

Reviewer #5

(Remarks to the Author)

(Remarks on code availability)

Version 1:

Reviewer comments:

Reviewer #1

(Remarks to the Author)

The authors have performed thorough revisions in response to the feedback from their paper's many reviewers, including multiple new analyses and figure panels. The careful edits (and patient descriptions in the response document) were very useful. The paper is useful and interesting, and I have no further feedback that should preclude publication.

(Remarks on code availability)

The analysis scripts for the paper are straightforward and well-commented. The repositories, plus associated wikis and website, are exemplary.

Reviewer #2

(Remarks to the Author)

We thank the authors for the changes made. We consider that both the manuscript and the pipeline have improved significantly, and that our initial comments have been adequately addressed.

There are two points we would still like to highlight:

- 1) The filtering of the studies (and its effects). It is surprising that the initial ENA search identified 255 studies, yet only 23 passed the filter—less than 10%. The remaining studies were imported from the HMC, which is also unexpected given that the HMC itself is based on an ENA search. It is therefore unclear why these studies did not appear in the initial search. It may be worth discussing what prevents the majority of studies from being usable in a meta-analysis. Is it the lack of metadata? Errors or inconsistencies? We believe it would be valuable for the manuscript to highlight this issue, as it emphasizes the need for improved study reusability.
- 2) Although some actions have already been taken in the discussion, we still feel they fall short. In particular, when reviewing the alpha and beta diversity analyses, the figures and data are typically labeled by country. This presentation can easily lead readers to assume that the results are representative of entire countries, despite clear biases in the underlying data. As mentioned in the manuscript, the high diversity observed in Peru is likely due to the fact that most samples are from the Amazon region, while the low diversity in Argentina likely reflects that all studies were conducted in industrialized settings. We suggest explicitly showing, in the figures, the number of studies and samples per country to help readers better assess the representativeness of the results.

One minor comments:

- 1) Just out of curiosity: if "South America" is always capitalized, why is the "SA" in saMBA lowercase?

(Remarks on code availability)

We thank the authors for adding an additional layer of reproducibility and containerization to the pipeline using Docker. While this introduces some complexity for users—particularly when verifying the Docker image and running it interactively—we recognize this as a valuable addition that functions well overall. However, we recommend reviewing the wget commands used for file downloads, as they may be slower than expected or prone to failure. This could be related to the limited resources typically allocated to the default Docker image, which may differ from the environment used during development

Reviewer #3

(Remarks to the Author)

The rebuttal letter and modifications to the manuscript adequately address all my concerns and comments.

(Remarks on code availability)

We appreciate the decision to follow our suggestion of a Docker container.

As well as the complete rewrite of the documentation to reflect this change — it's very thorough.

The installation is extremely simple now and takes just a few minutes.

We were able to run the demo completely, move the files to our machine and look through them.

We were also able to set up a docker volume and rerun the demo with both the input and output folders mapped to our local machine.

Additionally, the pipeline seems to be executable as well using the `input_samba_v0.1.0.txt` available on Zenodo. We did not run it fully, however, because of time and disk space constraints.

The scripts on the saMBA-article repository seem to be thorough, and comments on the code itself are generally clear.

It's unfortunate that the files on both this second repository and Zenodo do not include all the files needed to run the scripts without any modifications — the available metadata files seem to be missing or in a different format than needed. Still, we were able to replicate Figures 2b and c from the count table — so at least there's some confirmation that the figures can be reproduced from the count table.

Reviewer #4

(Remarks to the Author)

(Remarks on code availability)

Reviewer #5

(Remarks to the Author)

(Remarks on code availability)

We thank all the Reviewers for their thoughtful reading of our article and their constructive comments. We are grateful for the overall positive reception of our work especially in terms of its relevance and for acknowledging our efforts in making this article (and the resources associated) open to the community.

Below we outline our point-by-point response to the comments. In addition, we have updated the figures, and new supplementary material was added. We believe changes made to the current draft substantially improved the quality of our work.

Reviewer #1 (Remarks to the Author):

Valderrama et al. describe here a new integration of 33 microbiome projects from South and Central America. They describe the content of the samples and patterns at the country and continent levels, and discuss the implications for underrepresentation of populations from these regions relative to the rest of the world. Their approach is reasonable, consistent, and clearly described. Their analysis code is publicly available, and their pipeline is well-documented. They describe several interesting conclusions, and their model provides an example (and plenty of computational resources) for others to do similar work focused on other populations. I believe this is useful work that merits publication, pending necessary revisions to the text and possibly analysis.

Thank you for recognizing the impact of the work and the effort put into making our work open to the research community. We would just like to clarify a minor point made by reviewer 1, which is that we limited our search space to South America. Thus, Central America was not included.

Their results are mostly descriptive, which is not necessarily a negative factor. However, much of their justification for the work, as stated in the introduction and discussion, is in how the dataset compares to the Human Microbiome Compendium (HMC). The paper perceptively notes that the compendium's sample-filtering process excluded many samples from the region, but this manuscript would be stronger with a more quantitative consideration of these differences and a consideration of the implications.

For example, the paper notes that the HMC's exclusion of projects with fewer than 50 samples "unintentionally impacts world regions in a way that exacerbates existing differences in sampling efforts, as poorer regions may have higher proportion [sic] of projects with less than 50 sequenced samples." This would be a compelling finding, but it appears to be a guess, and according to Figure 1B, most of the projects included in saMBA also include more than 50 samples. It's not critical that this specifically be investigated, but it is one example of the manuscript raising questions that it does not address in its analyses.

Thank you for highlighting this important point. Although we determined that ~30% of the projects sequenced less than 50 samples in South America, doing so for other world regions would require applying the same manual curation we describe for saMBA to each of them, which is beyond the scope of this article.

However, we have reanalyzed the data available in the HMC to better support our claim that applying a threshold of 50 samples per sequencing project can “unintentionally impact world regions in a way that exacerbates existing differences in sampling efforts”. Please refer to **Suppl. Figure 2** in the revised draft. Please note that we are aware that the threshold of 50 samples was already applied to the data provided by the HMC, which limits our potential to directly assess that claim as those datasets are excluded already. Nevertheless, we believe the reanalysis still provides valuable insights that might support it.

We analysed the number of samples per sequencing project (panel A), and the number of samples per country (panel B) for each world region included in the HMC. China and the United States of America were removed from panel B, as both had sample counts some orders of magnitude higher than the next most-sampled country. Panel A and B show a vertical black line which represents the median number of samples in each case. Red dots represent projects (in panel A) and countries (in panel B) where the number of samples is above the global median. Thus, the reanalysis shows that ~22% of the microbiome projects conducted in Latin America and the Caribbean sequenced more samples than the global median, the lowest percentage across world regions. Similar patterns are observed in other world regions of Low- to Middle- Income Countries, such as Eastern and South-Eastern Asia (~36%) and Northern Africa and Western Asia (~43%). However, Sub-Saharan Africa, another region with many Low- to Middle- Income Countries, showed ~58% of projects with more sequenced samples than the global median, which is higher than the 53% of Europe and Northern America, and the 45% of Australia and New Zealand.

Additionally, we compared the number of samples sequenced in each country (panel B) against the global median. Interestingly, no country in Latin America and the Caribbean has more sequenced samples than the global median (0%). This pattern is observed in other world regions of Low- to Middle- Income Countries, such as Sub-Saharan Africa (23%) and Northern Africa and Western Asia

(25%). Contrastingly, the percentage of projects that sequenced more subjects than the global median raises in world regions with High Income Countries, such as Australia and New Zealand (100%) and Europe and Northern America (68%). Although this pattern is reverted in Eastern and South-Eastern Asia (67%), it is interesting to note that half of the countries in that world region are just in the boundaries of the global median.

Thus, the data from the HMC suggests that world regions with lower income tend to show lower numbers of samples per projects, and that countries of those regions tend to have less microbiome samples than High Income Countries. This is especially clear for Latin America and the Caribbean, further highlighting the relevance of our work.

This suggests that using thresholds of the number of samples sequenced as inclusion criteria may inadvertently exacerbate existing differences in sampling efforts. Thus, we believe this justifies the decision to create new local archives of gut microbiomes without using the size of the project as an exclusion criterion.

MAJOR NOTES:

Line 23–25: The paper states that this dataset is meant to address a sampling disparity that currently "limits generalizability and reduces the performance of predictive models." This may be intended as background information, but if a problem like that is stated in the abstract, it suggests this dataset will improve the performance of predictive models. This is not addressed in the results. In short, what are the implications of this data being available now? What analyses does it improve, and what could it be used for? Is there any analysis that would demonstrate this? We thank the reviewer for commenting on this which on reflection is a good point. The sentence was indeed intended to provide background regarding biomedical consequences potentially rising from the current imbalance in microbiome sampling across world regions. We have now removed it. Instead, we highlight more the fact that the disparity we tried to address in this article is the neglect of some populations rather than the lack of generalizability of predictive models.

Lines 153–162: This is a clear and useful description of the paper's filtering protocol, but it would be very helpful if it also described the effects of these steps, particularly because these steps are compared so directly to filtering from the HMC paper. The paper notes that the filtering has a more stringent effect on the smaller saMBA dataset, but I'm concerned it's having a much more dramatic effect than it might otherwise appear. For example, when filtering out taxa with fewer than 1000 reads across all samples, the HMC paper states that 50 percent of taxa were removed. The HMC dataset has about 57 times as many samples as saMBA and a higher median reads per sample, so I would guess the proportion of removed taxa would be far higher—1000 reads seems like it would be easier to find in the HMC's (very roughly) 6.9 billion reads compared to saMBA's 0.4 billion reads. So if the HMC analysis was based on the top 50 percent most prevalent taxa, what would that number be for saMBA? Even when evaluating a single dataset, I would guess results from the 5 percent most prevalent taxa would look different than the results from the 50 percent most prevalent. I may be

thinking about this the wrong way, but if the goal is to match the filtering done in the two datasets, I think it would be more appropriate to set the cutoffs to a level that achieves a similar proportion, rather than using the same cutoff.

We thank the reviewer for commenting on this important aspect of the analysis conducted in our research. We want to first clarify an error made in the reporting:

In the submitted draft, we reported the number of unique taxa found across all samples of saMBA to be: 2,246. This is corroborated by **Figure 2A**, by summing the number of taxa only found in saMBA, and those found in both saMBA and the HMC (1370 + 876). However, on re-examination there was an error when reporting the filtering process described in the section ‘Building saMBA’ (see methods). We mentioned that we used the same thresholds as the HMC, which we initially did. However, as noted by the reviewer, these thresholds were extremely stringent, as only 296 taxa (~13% of total) across the >2,900 samples passed the filter. Thus, when working on the analysis shown in the submitted manuscript, we used another set of thresholds to be more flexible, despite mistakenly keeping the text describing the original filtering strategy. This can be noted as, in **Figure 3A** from the submitted draft, we state that across all South American samples we found >1,200 different taxa, which would be impossible if we were working with a count table filtered using the approach described in the methods, where only 296 successfully passed all filters.

In this revised version, this error in reporting is fixed. To enhance the clarity and transparency of our work, we have further adjusted the thresholds to filter a proportion of taxa as close as possible to that used in the HMC. Therefore, we have made the following adjustments:

- 1) We have changed the filters steps used to build the ‘clean genus count table’ to match the proportions of reads removed from the count table at each step, as close as possible to what was reported in the HMC article (see methods).
- 2) In addition to stating the thresholds used on each filtering step in the methods section, we added the numbers of taxa and samples ruled out by each step. We added this new information hoping to improve transparency and to make it easier for readers to understand the effects of each filtering step.
- 3) We also updated the code in the GitHub repository to reflect the use of these filters (https://github.com/Benjamin-Valderrama/saMBA-pipeline/blob/main/scripts/misc/cleanup_archive_genus_table.R).
- 4) Finally, we want to clarify that changes in the filtering process would only affect **Figure 3** and **Suppl. Figure 5** in the revised draft (other figures used the unfiltered count table). In both cases, the figures were made again to account for these changes and due to other comments made by reviewers. Note that the new Figures don’t alter the conclusions of our work.

Lines 197–208: I may be misinterpreting the method here, but I believe the approach as described does not account for sampling effort, which could be artificially inflating the richness of individual countries. For example, we see that at 100 samples, the number of novel taxa observed in Ecuador is (on average) much lower than the number observed in Peru. The number of samples from each

country is identical in this comparison, but if the projects in Peru sequenced their samples with, say, five times as many reads, then it may be misleading to say that the microbiomes of one region harbor more novel taxa than the other—one of those countries was just captured at much higher resolution. I believe one way to mitigate this would be at least similar to the approach taken in the referenced HMC paper—samples were randomly rarefied to 1000 reads each prior to the sampling process described here. A chart using this data may look very different, but it would be comparing countries on more even footing: If you took, 1000 reads from Peru and 1000 from Ecuador, which would contain more novel taxa? It doesn't look like the HMC analysis randomized the samples for every iteration, but the main point is simply that it could be much more interpretable to account for number of reads per sample here.

I realize that could be a computationally intensive effort; one alternative, but not the only one, could be to just interpret this plot slightly differently by saying it assumes disparities in reads per sample will continue as they currently are.

We thank the reviewer for the comments on the implementation of the subsampling strategy to describe the extent to which biodiversity across the world region is known. First, we would like to mention that our method follows the same principle as described for **Figure 4A** in the HMC article (see text under title “World region taxon discovery rate” in the methods section of the article [https://www.cell.com/cell/fulltext/S0092-8674\(24\)01430-2#sec-8](https://www.cell.com/cell/fulltext/S0092-8674(24)01430-2#sec-8)). Authors took a N number of samples from each world region and recorded the number of unique taxa present across those N samples. A process which was repeated 1,000 times for each sample size. Thus, they didn't rarefy to account for sampling effort when conducting that analysis.

However, attending to the comments of the reviewer, we have added a new **Suppl. Figure 5** (see figure below and text in the revised manuscript for details). There, we took an increasing number of samples for each South American country, rarefied the number of reads of each sample and determined the number of unique taxa. We performed the analysis twice, rarefying to 9,000 and 1,000 reads (90% and 10% of the sample with the lowest number of reads in the entire saMBA dataset, respectively). For each number of samples and rarefaction regimen, this process was repeated 1,000 times, and the mean number of unique taxa was used for plotting. This process was repeated using the filtered version of the saMBA dataset (after removing rare taxa and samples, as described in the methods) and the dataset before applying any filters.

Since the results generated without rarefaction and the two rarefaction regimes produced similar results, we have kept the version without rarefaction in the main **Figure 3A** and added the panels with the other iterations as part of the **Suppl. Figure 5**.

Lines 260–262: The statement here reads, "the exclusion criteria used in building the HMC increases the underrepresentation of South America, and potentially the global south." This may be true, but it appears to be unsupported by the results, at least as presented. Though "30% of the studies included had less than 50 samples," this says nothing about the proportion relative to other regions. It seems to me that "underrepresentation" would only happen if other world regions have a differing distribution of samples per study: If 30% of South American studies have less than 50 samples, and, say, 60% of studies from Europe, a 50-sample cutoff would inflate South America's representation, not the other way around. This is a compelling question, but if it's included here, it should be better supported.

Thank you for commenting on this important point. We want to highlight panel A of the newly incorporated **Suppl. Figure 2**, where the distribution of sample sizes per study is depicted. There, we showed that only ~22% of microbiome projects from Latin America and the Caribbean included in the HMC have more than the global median number of samples per project. Contrastingly, this proportion goes up to ~54% for the region of Europe and Northern America.

As mentioned above, we restate that we are aware that the threshold of 50 samples was applied to the data provided by the HMC, which limits our potential to directly assess the quoted claim.

Nevertheless, we believe this reanalysis still provides valuable insights, as it suggests that Latin America and the Caribbean tend to have smaller number of samples per microbiome project, thus justifying a regional screening without considering the number of samples per project as an exclusion criterion to improve the representativity of the continent.

Lines 291–294: The use of alpha diversity measures here, particularly Chao1, seems at odds with the paper's description of the filtering of low-abundance and low-prevalence taxa. It seems filtering out those taxa would disrupt the relationship between the two measures presented. In addition, there have been concerns raised (DOI 10.1093/ismejo/wrae106) about the Chao1 estimator being used not just for filtered data such as this, but for some amplicon analyses in general. In short, if Chao1 is attempting to estimate unseen taxa, but many have already been filtered out, it's unclear what this might indicate; the paper would benefit at least from some elaboration here.

We thank the reviewer for highlighting potential issues in estimating unseen taxa (with Chao1) from a pre-filtered count table where low abundant taxa were discarded. We would like to highlight that in the methods section we stated that when comparing saMBA to the HMC, the unfiltered count tables were used. To address potential confusions, we listed all the figures where the unfiltered table was used for the analysis, as stated now in the corrected version.

Regarding the article mentioned by Reviewer 1, it suggested that Chao1 should be avoided when working with singleton-removed ASV data and suggested the use of OTUs instead. We prefer working with ASVs instead of OTUs for the same reasons the article states as strengths of the former over the latter. Importantly, note that we used genus-level (not ASV-level) count tables to calculate the alpha diversity indices shown in all panels of **Figure 2**. Regardless, to incorporate the comments of the reviewer, we have now used Observed genera instead of Chao1 through the entire manuscript. Hence, all mentions to Chao1 were thus changed, and **Figure 2** in the revised manuscript was updated accordingly.

Lines 311–312: I am a little confused by the logic in the paper's interpretation of Figure 3A, which states that "although some countries are close to reach [sic] a plateau, none has done so yet, suggesting that newer samples taken in most South American country [sic] will likely identify new genera not yet included in saMBA or any other compendium." It's unclear to me how this conclusion is reached, particularly because the "continental estimate" in Figure 3A has plateaued. If the lines for each country continue to rise, but the pooled results ("South-america" in 3A) are no longer growing, this could also suggest that most South American taxa have been observed (hence the flat black line), but that those taxa are present at different prevalences in each country. If this were the case, then it may be that further sampling from Bolivia would reveal a taxa novel to Bolivia, but that would already have been observed in multiple other countries. Even if the continental estimate had not leveled off, however, it's not clear here why a taxon novel to South America would "likely" be unobserved "in any other compendia." It seems possible, but if there are unidentified taxa in a region, how would one discern the probability that those taxa have not already been observed elsewhere?

We thank the reviewer for this thoughtful comment regarding potential misinterpretation of the results. We agree with your interpretation, as taxa not found in one country may have still be found in others, thus not adding novel taxa to the entire continent. We have changed that accordingly to the manuscript. In addition, we have repeated the subsampling analysis, this time using the unfiltered saMBA dataset (without removing rare taxa, as stated in the methods). In the unfiltered dataset, all countries (and the continent) shown a constant increase in the number of unique taxa identified as more samples are included in the analysis. This new result is consistent with the conclusion we reported in the submitted version of our manuscript. Hence, we have changed the text to describe the results and conclusions extracted from these two analyses. Please see the revised text of **Figure 3**.

Line 463: This section states that "world regions used in previous work are too broad," but it is unclear what this means—"too broad" for what? The fascinating result from Supplementary Figure 1 supports the statement that "explorations at a finer geographic resolution are justified," but if the criteria for justifying these explorations is that between-group differences can be identified, this chain could continue down the line: maybe continental regions are too broad because they fail to account for urban vs rural differences, and those classifications could be too broad because of differences between landlocked cities and those on the coast. If the regional level is "too broad" but the level presented here is correct, the manuscript would benefit from an explanation of why this specifically is the correct level.

We fully agree with the logic of this comment, and as reviewers 1 and 3 highlighted, we are aware that even more restrictive search spaces are equally —or potentially more— meaningful than continents for certain scientific questions. This is especially relevant where environmental and cultural factors occur in gradients within close geographic proximity, as we highlighted later in the submitted manuscript. In the text, we describe contrasting lifestyles in Brazilian cities and communities within the Amazon, which was further expanded in the text of **Suppl. Figure 6** of the updated draft.

We thank the reviewer for highlighting this sentence, as it was indeed not intended to convey the idea that continents are the ideal search space. Instead, our intent was to suggest that performing the analysis at the level of global regions, as used in the HMC, is probably too coarse. As reviewer 1 mentioned, we showed that in our **Suppl. Figure 1**, where we identified a visible distinction between Central and South America, despite being unified into the category "Latin America and the Caribbean" in the HMC. This information was used then to justify the restriction of the search space to South America, which is a finer (and potentially more biologically/culturally relevant) level of resolution than Latin America and the Caribbean, but still with potential to be of interest to a broader community of researchers and readers.

To avoid potential misinterpretations, we changed the phrasing to remove emphasis on the word *continent*.

MINOR NOTES:

Line 140: This notes that the "v1.30.0" version of DADA2 was used here, but it appears the most recent version of that software is 1.26.

We thank reviewer 1 for highlighting this. We revisited the yaml file with the description of all dependencies installed in the environment used for the analysis conducted in this article. We corroborated that DADA2 v1.30.0 is the version installed and used.

It seems that v1.26 is the latest version uploaded to the GitHub repository in November 2022 (see here). However, the software has been under active development ever since. The package is now primarily maintained in Bioconductor, where the latest version, v1.36, is available for R v3.5.0 (see here) since the 14th of May 2025 (see DADA2 reference manual here). Thus, we consider it appropriate to keep the version stated in the manuscript as it currently is.

Line 224: It would be helpful if the statement about pathogen/parasite exposure's effect on the microbiome was supported by a reference.

We have now added references to substantiate the claim.

Lines 155–157: I was confused by the statement here that "the HMC remove taxa present in less than 1,000 samples," because it doesn't seem to align with the filters in the previous sentence (taxa with fewer than 1,000 reads across samples, or taxa present in less than 100 samples). It also doesn't appear to describe the HMC filtering referenced: "To reduce sparsity introduced by exceedingly rare taxa, we then removed 2018 taxonomic entries (50%) with fewer than 1000 total reads across all remaining samples, and a further 578 taxa (14% of the original total) that were detected in fewer than 100 samples." The manuscript would benefit from clarification here. Thank you for spotting this error in the reporting. Please note that due to one of the major comments Reviewer 1 made above, that section was rewritten (see above). In the current version this mistake was corrected.

Lines 185–195: The Local Representation Index is a useful tool here, but the description as written is difficult to follow. Phrases like "sample percentage" and "how many times the number of samples needs to be increased" could benefit from rephrasing.

We have changed the description of the method as suggested. Please see the revised version of the manuscript.

Line 263: This states that 9 of 13 countries were present in the dataset—does this refer to the filtered or unfiltered data? If the former, it would be helpful to know if other countries were present but got filtered out.

Thank you for flagging this. Although samples were discarded for some countries, the process didn't remove all samples from any country. We have clarified that in the text where the workflow is described as shown below. Please see the text in **Suppl. Figure 3** in the revised manuscript.

Line 264: I believe this is the only point in the text where the number of samples is specified. This is an important enough number to put in the abstract, not just on the bottom of page 7.

We thank the reviewer for acknowledging the relevance of the number of samples included. We have added that information in the abstract.

Supplementary Figure 3C: If this plot was made using a dissimilarity matrix, would it be PCA (principal component analysis), as stated, or PCoA (principal coordinates analysis)? We agree that PCoA is the appropriate name for the analysis described, as we calculated Bray-Curtis dissimilarities to generate the figure. We have now corrected the text.

Reviewer #1 (Remarks on code availability):

The analysis scripts for the paper are straightforward and well-commented. I wasn't able to find the code used for filtering samples and taxa, but this is likely user error on my part. The repositories, and associated wikis and website, are exemplary.

We thank Reviewer 1 for their comments regarding our efforts to make this article and its associated resources open to the community. The script was available in the GitHub repository linked in the manuscript. Here we included a link in that script within the repository folders (see here).

Reviewer #2 (Remarks to the Author):

The manuscript by Valderrama et al. aims to generate a resource focused on highlighting studies related to the South American Human Gut microbiome. This resource is intended to facilitate comparative analysis while implementing open-access, reproducible bioinformatic pipelines.

The work is of significant interest, as it has been reported that Latin American data are underrepresented in human gut microbiome studies. The manuscript adequately emphasizes that the most recent reports, while applying filters based on the number of samples per study to improve computational efficiency, simultaneously introduce bias against studies with smaller sample sizes—an issue particularly relevant to regions with limited resources.

My main concern relates to the search, selection, and filtration of studies and samples used in the final analysis:

Although I agree that this is the largest compendium of microbiome studies from South America to date, it does not constitute a comprehensive collection of all studies that have published South American microbiome data, and this limitation should be explicitly acknowledged. Furthermore, while the effort to ensure transparency and reproducibility in all steps is commendable, there are still improvements that could be made in this regard.

For example, it would be valuable to list the raw results from the initial search and to show how successive filtration steps reduced the number of studies, leading to the final list presented in

Supplementary Table 1. A representation similar to Supplementary Figure 2 could be expanded to include the list of studies and all filtering steps, both prior (e.g., sequence type) and subsequent (e.g., abundance, prevalence).

Thank you for your enthusiastic and constructive comments on the paper. We agree that our method of screening the INSDC through the ENA portal potentially limits our ability to find publicly available human microbiome data from South America in cases where the sample metadata or the information provided by the original authors doesn't explicitly acknowledge the geographic information. This is indeed a limitation, and it is unfortunately shared with other compendiums (like the HMC) using automated or semi-automated searches. For instance, in the HMC they acknowledge that (2.7%) of the included samples come from other body sites or from non-human sources due to poor metadata annotation made by original authors when uploading their sequences to the INSDC. Of note, all samples included in saMBA were manually curated, and that is not a limitation in our work. To address the comments of the Reviewer; we have now explicitly acknowledged that in the Limitations section

However, we believe that despite the above limitation, our screening effort was effective in retrieving all identifiable South American gut microbiome samples present in the three major repositories of biological sequences (ENA, NCBI and DDBJ).

Regarding the suggestion of building a figure showing the exclusion of projects, we originally decided to not do so as it is composed of a one-step decision: it can either be included or not. The inclusion criteria used were explained in the methods section of our manuscript. We have now added a **Suppl. Table 2** where the number of discovered and discarded projects for each country is stated. In line 78, the manuscript states that the authors "screened the INSDC" and manually curated a list of 33 gut microbiome studies. However, Supplementary Table 1 shows that some studies were retrieved from ENA, while others came from Google Scholar or the Human Microbiome Compendium. We thank the reviewer for highlighting any potential confusions generated by our phrasing. We screened the INSDC as stated in the introduction and further explained in the methods section as follows: "The European Nucleotide Archive (ENA) was used as an interface to systematically search the INSDC."

We would like to clarify that The International Nucleotide Sequence Database Collaboration (INSDC) is a "global collaboration independent of governmental or non-profit organizations that manage nucleotide sequence databases (...) to create a comprehensive collection that preserves the scientific record and enables broad sharing of such data." The members of this global initiative are the European Nucleotide Archive (ENA), the National Centre for Biotechnology Information (NCBI), from the United States, and the DNA Data Bank of Japan (DDBJ). In their website they mention that each member "Exchanges data with other members at regular intervals at each site and sufficient redundancy in case of a catastrophic failure". Thus, data originally submitted to the NCBI or DDBJ can be found through the ENA search engine soon after. Authors conducting the screening (BV and PC-R) thought ENA provided an interface that was more user-friendly than the other two INSDC members, leading to choosing that platform to work with.

An additional filtration step states that “samples with more than 10% of reads assigned to Archaeal taxa or with more than 10% of reads unclassified at the phylum level were also discarded.” The rationale behind this filtering criterion should be clearly explained. Similarly, the manuscript states: “Second, projects with 5 or more of the first 10 samples with >25% of chimeric sequences were also discarded.” It would be helpful to justify why a project-level filter was applied here rather than a sample-wise filtering approach.

The original removal of samples with more than 10% of the reads assigned to Archaeal taxa was a mistake originated from a misinterpretation of the code shared by authors of the HMC. We rigorously tried to replicate the analytical framework used in the HMC, so we didn’t just limit the replication effort to the methods section, but we also looked at the source code, which authors of the HMC have now archived outside GitHub (see here). In the code they generated a variable to remove samples with >10% of reads assigned to archaea. However, they didn’t use it later in the code. Although no samples were removed due to this filter, this step was removed in the current code used in saMBA and uploaded to GitHub.

The second criterion, which was applied at the level of project instead of single samples, also follow the guidelines described by the HMC article. Although not explicitly stated by authors of the HMC in this context, this is what they mention when discussing their approach to reanalyse paired-end sequencing projects as single-end when less than 50% of forwards reads merging successfully:

“We believe this was the most reliable way to process this data, given the lack of information about project-level sequencing strategies. The merging process in paired-end datasets would be much more effective with more knowledge of study design, particularly in cases where the amplicon length was greater than the read length and the paired-end reads did not overlap. In addition, DADA2 recommends building separate error models for each sequencing run, but only BioProject could be reliably inferred, which means ASV inference at the study level may not capture run-level patterns.”

Suboptimal determination of error rates by the DADA2 algorithm may lead to poor merging and unexpectedly higher levels of chimeric reads. Thus, we believe project-level filters were applied to further ensure reliability on the results generated in the analysis of each project. We, thus, used the same thresholds to account for the same potential issue. We have now explicitly stated that in our methods section.

Other concerns:

1. The manuscript does not address potential batch effects in the analysis. The Human Microbiome Compendium (HMC) claims that batch effect correction is unnecessary due to the large number of studies analyzed, which allows biological variation to dominate. However, in this work, the number of studies is more limited. Therefore, the potential need for batch or study effect normalization should be discussed.

We thank the reviewer for commenting on the potential effects of not accounting for batch effects in our analyses. We believe that although batch effects were important to consider for the authors of the HMC, it is less so for saMBA. In the HMC, authors run different statistical inference test and machine learning algorithms to test hypotheses. Two examples are that specific genera will be differentially abundant across world regions, or when they trained ML classifiers to identify the world region samples came from. In such cases, reducing technical variability to enhance the ability of the models to correctly learn from the biological signatures in the data is critical.

In contrast, in this article we only perform statistical inference when comparing the HMC against saMBA. However, when we do so (**Suppl. Figure 4**) we are interested in both sources of variability: technically- and biologically driven. We believe that accounting for batch effects would be detrimental to our assessments.

2. The selection of beta-diversity metrics requires clearer justification. While Jaccard dissimilarity is used in one section, Bray–Curtis dissimilarity is employed for comparison with HMC results. High within-country Jaccard distances are interpreted as indicative of microbiome uniqueness; however, this metric is sensitive to rare taxa, which may be prevalent in several of the included studies. These values can be influenced by sequencing depth, compositional sparsity, and inter-study technical variability. These factors should be considered in the discussion.

Thank you for commenting on our decision to use different beta diversity metrics. We used Jaccard dissimilarity to explore the uniqueness of the microbiomes within each country (**Figure 3C**), as we wanted to assess if different microbiome samples share the same bacterial members. Note that this analysis was performed after applying the filtering steps that removed rare taxa. Thus, extremely rare taxa were removed and can't skew our results.

On the other hand, we used Bray-Curtis dissimilarity (**Suppl. Figure 4**) because it not only considers the presence (as Jaccard does) but also the abundance of the taxa within each sample. This second aspect is critical to assess the similarity of the community profiles produced by both workflows tested: the one developed by the authors of the HMC and the one we built.

3. The use of the Local Representation Index (LRI) is interesting, but it may assume that sampling is random across the population, which is not the case for any of the included studies.

We agree on that current sampling efforts may not be random, and we extensively discussed the internal differences within South American countries, advocating for the future use of finer scales of the analysis at the country level (lines 430-432 of first draft). To address the comments made by the Reviewer 2, we added lines 547-549.

4. In the comparison between different workflows, there are established metrics, such as Procrustes analysis or mantel tests, to assess whether beta-diversity profiles are equivalent. It would be

informative to investigate the samples with Bray–Curtis dissimilarity greater than 0.25 between workflows. Was there any pattern associated with these discrepancies?

We thank the Reviewer for this thoughtful suggestion. We have conducted a Mantel test to assess the correlation between the pairwise dissimilarity matrix of the HMC and that of saMBA, and added the results to the updated manuscript. See the revised text associated with **Suppl Figure 4**:

The Mantel statistic ($r = 0.9813$, $p = 9.99 \times 10^{-5}$, 10,000 permutations) indicated a very strong and statistically significant correlation between the two matrices. This result confirms that our workflow preserves the core structure of microbial community dissimilarities observed in the HMC workflow. Consequently, it supports the compatibility of the two workflows, suggesting that results from saMBA can be integrated with those from the HMC workflow without additional normalization or preprocessing.

5. In the analysis of non-industrialized settings, although the number of samples is reported (1,072 vs. 446), the number of studies or sampling sites is not provided. This information could significantly affect the interpretation of the results and should be clarified.

We have added this information to the manuscript. Regarding sampling sites, we have added that information to **Suppl. Table 1**. Regarding the comments made by the Reviewer: 4 projects included in this analysis sequenced gut microbiome samples from individuals in the Amazon. The other 4 did so from individuals living in small villages that were in mountainous areas or with limited interaction with other populations. For instance, in one of the articles, authors reported that sampling sites were in places hours away from the nearest food market, so sampled individuals often owned hunting or fishing items to procure their own food.

6. The terms “metagenome,” “metabarcoding,” and “shotgun metagenomics” are used interchangeably throughout the manuscript, which may lead to confusion. Defining these terms early in the text would improve clarity.

Although we tried to identify where in the text those words were used but unfortunately, we couldn't do it, neither by reading the manuscript again nor by using the word search tool provided by Microsoft Word and Adobe Acrobat. Moreover, our work only considers 16s rRNA data. Thus, we didn't change anything in the text that could address this comment.

7. The classification of “industrialization” requires further refinement. South America includes both highly industrialized urban centers and remote rural regions. However, the manuscript appears to generalize industrialization status at the country level. While classifying samples by industrialization can offer ecological insight, metadata quality and consistency vary widely across projects. In some cases, assumptions about rural versus urban settings may oversimplify complex lifestyle gradients.

We thank the reviewer for commenting on this important point. We agree that industrialization status can't be generalized to the level of countries. Indeed, we highlighted that very fact throughout the manuscript. See here as an example:

“Although these results may help in deciding where to sample next, we recognize some limitations arising from the high levels of heterogeneity among South American countries. For instance, while Sao Paulo city has around 44 million habitants, a minor yet often studied part of their population lives in the Brazilian amazon region, with close to zero contact with the industrialized world. It's expected that these populations will differ dramatically. Additionally, social inequities —that can shape our gut microbiomes in multiple ways— are prominent within South American countries like Chile.”

To assess if the microbiome projects included in saMBA profiled individuals living in industrial or non-industrial settings, we relied on information reported by authors either in the methods section, in the results section (when describing the cohorts), or in the metadata uploaded by authors to the INSDC. In many cases, assigning samples to industrial or non-industrial settings was straightforward. For instance, many projects state they recruited individuals from cities with >10 million individuals, such as Santiago, Buenos Aires or Sao Paulo. Similarly, other projects stated they collected samples from small communities living in the Brazilian, Peruvian or Venezuelan sectors of the Amazon, or in small towns where the closes market is more than an hour to travel in motorized vehicles. In cases where such categorization was not as clear, we left those projects as 'NA' (see updated **Suppl. Table 1**, where we now added more information about sampling sites).

Finally, there was one project (PRJEB39990), where samples from industrialized individuals (living in a Brazilian city) and non-industrialized individuals (living in the Amazon) were uploaded together to the INSDC. For this project, we were able to determine the industrialization category using the metadata provided by the authors when uploading the fastq files to the INSDC in conjunction with the information provided in the methods section.

Minor comments:

1. Why is Figure 1B cited before Figure 1A?

We've now changed the order of the panels to match the flow of the text. Figure caption was changed accordingly.

2. Line 264: Given that the standard deviation is larger than the mean for the number of samples per study, it would be helpful to show the full range or distribution. We thank the reviewer for this suggestion. We have added that information to the text.

3. Line 290 states that the analysis “unveils a high biodiversity,” but without comparisons or references, it is difficult to assess whether the reported values are indeed high. We thank the reviewer for highlighting this issue regarding the lack of references to compare with. We have modified **Figure 2** to allow an easier comparison of the diversity indices for South America

(as calculated by the HMC and saMBA), and to compare South America (from saMBA) to the rest of the world (from the HMC).

4. Line 311: "South American country" should be "South American countries." The error was corrected. Thanks for spotting this.

5. Figure 3A: The colors used for Bolivia and Venezuela are difficult to distinguish. Consider improving the color contrast.

Thanks for pointing this out. We have now changed the colour of Bolivia and kept the orange for Venezuela. We are aware that Bolivia, due to the low number of samples sequenced, is almost entirely covered by the other countries. We tried with different colours and by forcing Bolivia to be plotted on top of the other countries, but none really solved the issue.

6. Lines 94–97: The search strategy is clearly described, but it lacks reproducibility. Consider including the exact search strings used for each country and the date(s) of access to the ENA database.

We thank the reviewer for highlighting potential issues with the reproducibility of our work. We would like to mention that the search term and dates of access were already provided in the **Suppl. Table 1** in the columns called "Search term" and "Date accessed".

7. Lines 106–115: It is unclear whether samples with ambiguous metadata (e.g., uncertain geographic origin) were included or excluded. Please clarify the exclusion criteria related to metadata uncertainty.

Thank you for commenting on this. Samples with uncertain geographic location were not included in saMBA. We previously mentioned that samples were "limited" to those fulfilling the three criteria described in the methods. To avoid confusions, we now added: "Samples without clear information regarding those three criteria were discarded from the analysis"

Additionally, we explicitly mentioned something similar regarding full projects under "Identification of bioprojects included in saMBA"

8. Lines 219–225: Clarify how disease and non-disease samples were separated. Was this distinction based on project metadata or manually extracted from publication abstracts? Also, indicate whether any filtering was performed for co-morbidities. We thank the reviewer for commenting on this matter. No filtering was applied for comorbidities. Samples provided by subjects with disease can't be separated from those without in many studies due to limited metadata availability or consistency in that regard. Thus, the column "included_disease" from Supplementary Table 1 contains information on whether the study included samples from individuals with disease or not. We have now explicitly mentioned that in the **Suppl. Figure 6**

Reviewer #2 (Remarks on code availability):

Comments on the saMBA pipeline (<https://github.com/Benjamin-Valderrama/saMBA-pipeline/wiki>):

1. The command to create the environment uses an outdated file name: `micromamba env create --name samba --file env/samba.yaml`
This should now reference `samba.yaml`.

Please see the response below, to point 3.

2. The current environment specification may lead to dependency resolution issues: Could not solve for environment specs
The following packages are incompatible:
└─ _libgcc_mutex ==0.1 conda_forge does not exist (perhaps a typo or a missing channel);
└─ _openmp_mutex ==4.5 2_gnu does not exist (perhaps a typo or a missing channel);
└─ binutils_impl_linux-64 ==2.40 ha1999f0_7 does not exist (perhaps a typo or a missing channel);
This suggests that the `samba.yaml` file in the repository should be updated. It would help to refresh the environment file using more stable channel instructions (e.g., include `noarch`, review Bioconda priorities, and ensure compatibility with `conda-forge`). It should also be noted that the latest `mamba` solver (v2.0.0) introduced significant changes, which have caused many Bioconda packages to fail. A warning or guidance for users on this point would be useful.

Please see the response below, to point 3.

3. The current environment file includes 324 dependencies. Reducing the number of packages, where possible, would improve reproducibility and reduce potential for conflicts. We thank reviewers for their effort in trying to replicate the analysis. We have created the environment file from scratch installing only the following software (note that micromamba installed the required dependencies automatically):

```
$ micromamba install python=3.10.0 bioconda::fastq-dl=2.0.4
```

```
$ micromamba install conda-forge::r-base=4.3.3
```

```
$ micromamba install bioconda::bioconductor-dada2=1.30.0
```

```
$ micromamba install conda-forge::r-tidyverse=2.0.0
```

The updated file with the environment is now in GitHub.

In addition to this effort, and following the suggestion from Reviewers, we transitioned to Docker to ensure full reproducibility, and to facilitate the deployment of the workflow by third parties. The file to build the container is now shared in GitHub. The built image can be accessed from Docker Hub

and instructions to download and deploy the workflow in Docker can be found in the workflow documentation and the README.md file, where Docker is the recommended option.

4. The instructions mention that installation and environment setup may vary. This section should point to troubleshooting resources or include common alternatives to improve user support. The instructions to use our workflow now recommend Docker as the preferred option to enhance reproducibility. This reduces potential differences in the setup and drastically reduces the need for troubleshooting in the preparation stage, as everything needed to run the demo is already present in the Docker image.

We provided instructions to download the Docker image and deploy the software. We provide instructions to run a demo, where 27 of the >3,000 samples included in samba are reanalysed and consolidated into one output table. We also show how to retrieve that output from the Docker and host it in the local machine of users. Finally, we redirect users to Docker documentation on how to work with volumes, which will be required to run the workflow when building their own resources.

5. Consider making the reproducibility-related instructions more prominent, such as placing them in the main README. The Wiki tab is often overlooked in GitHub repositories. Thank you for highlighting this important point. Instructions to run the demo are now added to README.

Reviewer #3 (Remarks to the Author):

The overall focus of the work is to amend/expand the HMC compendium with a manually curated selection of south American microbiomes from the public repositories. This is of interest to the community as the region remained under-sampled in that compendium despite its previously observed high biodiversity. The work also provides valuable results that could guide future human gut microbiome sampling campaigns in south America, along with publicly available bioinformatic pipelines that can be applied to other continents in a similar fashion. On the other hand, the manuscript's claims that it "deepens our understanding of the different stable states of the human gut microbiome", "generated the most accurate assessment of regional biodiversity to date" or expanded "the concept of the healthy microbiome to be more globally representative" are not sufficiently substantiated by the results; the proposed expanded compendium could be used to that end indeed, but the analyses and scientific results to substantiate such sentences are not sufficiently present in the manuscript itself.

The presented analyses are sound, along with its interpretations. The bioinformatic methodology is adequate and mostly well described.

The code and datasets included in the repositories are well presented and clear. In fact, the authors' efforts towards transparency and open science are commendable, including the public availability

of the pipeline, the extensive documentation, and the invitation to external researchers to directly share new studies for their inclusion in the database. Nonetheless, we spent a considerable amount of time trying to install the saMBA pipeline on Linux, but ultimately we were unsuccessful. We're still unsure whether the issues stemmed from some incompatibility on our end, but we tried several troubleshooting steps: installing micromamba instead of using conda, removing conda's activation steps from the ~/.bashrc file altogether, resetting the PYTHONPATH to an empty string, removing some possibly conflicting packages from our system, and even simplifying the .yml file by removing version constraints and all R packages. None of these attempts resolved the problem. Perhaps the pipeline runs smoothly on a freshly installed Linux system, but in our case we were unable to get it working. We also tried running the script without installing anything, but without fastq-dl, which we also failed to install, we weren't even able to download the test files. As other researchers could face similar issues, a possible option would be to provide a Docker container, which is fully self-contained and would ensure reproducibility and usability.

Thank you for recognizing the impact of the work and the effort put into making our work open to the research community. Following their suggestion, we transitioned to Docker to ensure full reproducibility. The file to build the container is now shared in GitHub. The built image can be downloaded from Docker Hub and instructions to make a local copy and to deploy the workflow in Docker can be found in the workflow documentation and the README file, where Docker is the recommended option for reproducibility.

The author's should check whether their research follows the journal's 'Sex and Gender Equity in Research – SAGER – guidelines'. For instance, i) they should mention that the sex and/or gender of the metadata likely arises from self-report, and ii) "Data should be reported disaggregated for sex and gender where this information has been collected". We thank the reviewer for highlighting this important consideration. We provided information regarding sex of stool samples donors in the nr-reporting summary sent by the editor. We have added part of that information to the text. See below:

"The sex of microbiome sample donors couldn't be reliably identified for all samples, so the information was not included in the analysis."

SPECIFIC COMMENTS:

Major comments:

L28. It would seem that the results provided on this subject are not related to other populations. Thus, what does the term "high" actually mean here if it is not properly compared to other studies? Are these results normal if compared to other global populations; are south-american microbiomes particularly rich and unique?

We thank the Reviewer for commenting on the lack of reference points for comparisons in **Figure 2**. We would like to highlight that we indeed compared the median value of the Shannon diversity index

obtained in saMBA to the same index reported in the HMC for Latin America and the Caribbean. This can be observed in lines 374-379 in the original draft submitted.

We also made a new version of **Figure 2**. Now the global distribution of the indices “Shannon diversity” and “observed genera” are shown for world regions covered in the HMC and saMBA. Note that we have used Observed genera instead of Chao1, as the use of the latter was questioned by one reviewer. Additionally, just for this figure, the region Latin America and the Caribbean (used in HMC) was divided into two subregions: (1) South America and (2) Central America and the Caribbean. This facilitates the comparison between the South American samples included in saMBA and in the HMC. See figure panels:

Note that results here shown were generated using the unfiltered HMC and saMBA count tables (which was stated in the methods section). Therefore, Shannon diversity values calculated for each region are slightly bigger than reported in the HMC article.

The reason to work with the unfiltered table is that we failed to replicate the process used by authors of the HMC to remove taxa. They state that the unfiltered table had 168,464 samples, and they then mentioned that 16,781 samples were removed, as they had less than 10,000 reads. Then, they removed 2018 taxa with less than 1,000 counts across samples, and then they applied a final filter to remove 578 taxa present in less than 100 and samples. Thus, at the end of the filtering process they ended up with 150,721 samples and 1422 taxa.

We downloaded the unfiltered count table using the R package they developed called MicroBioMap (note that the filtered table is not available). It had 168,464 samples, as reported in the original publication. Then, we were able to remove the 16,781 samples with less than 10,000 reads, as they reported. However, when removing taxa with less than 1,000 counts across samples, we removed 2,485 instead of the 2,018 taxa they reported. As we couldn't replicate their results, we decided to

work with the unfiltered count table provided by the HMC, and the unfiltered count table generated in our work for saMBA.

L29. “expanding the concept of the healthy microbiome to be more globally representative” I do not see this sentence as related to the work. Indeed, the curated database could be used to that end, but there are no particular related results on the matter within the manuscript. We thank the Reviewer for commenting on interesting future uses of this database. We have modified the sentence in the revised manuscript and have removed explicit reference to a healthy microbiome.

L33. “The framework used to build saMBA is compatible with existing global resources”. Further information on comparability (see comments below) and compatibility is required. For the latter, what do the authors mean by “compatible”, is it that one can download profiles from both resources and analyzed them together without extensive processing?, how?.

We have extended the assessment of compatibility using the Mantel Test as suggested by another reviewer. These changes can be seen in **Suppl. Figure 4** from the revised manuscript. See response also to Reviewer 2.

L81. Can you expand on the issue? What is available at HMC (e.g. filtered genus-level count tables, unfiltered ASV tables, fastq sequence and metadata)? What is not available with HMC? It is hard to identify which of the two sentences that are part of line 81 this comment is referring to. If related to the inclusion of new gut microbiome studies (first part of line 81), it was explained more extensively in the methods section, under the title “Identification of bioprojects included in saMBA”.

If referring to our effort to replicate the workflow described by authors of the HMC that unfortunately was not made available upon publication, then it is explained with more detail in the section “Analysis of individual projects included in saMBA”. Regarding this second part, we believe the effort made by the authors of the HMC in extensively describing what type of resources were made publicly available far exceeds what could be covered by a couple of sentences in this article. However, we have now rephrased the explanation of the recreation of their workflow to make it clearer.

L151-156. Where the genus-level tables, from which most analyses derive, normalized or subsampled in any way? (e.g. by subsampling to a fixed sequencing effort). Most analyses may be biased otherwise (e.g. Bolivian samples in fig3A may harbor more taxa than those of Ecuador due bias in the sampling depths of both projects, or the use Jaccard distances to estimate microbiome uniqueness within countries; it is unclear how "presence" was defined across samples with differing sequencing depths.)

We thank the reviewer for highlighting this important aspect of our analysis. We would like to highlight that in analysis where saMBA was compared to the HMC (**Figure 2**, **Suppl. Figure 4** and **Suppl. Figure 7**), the unfiltered count tables of both resources were used, as stated in the methods section of the corrected manuscript.

Regarding results in **Figure 3A**, the count table used to generate the plot in the original draft was filtered to remove rare taxa, as described in the updated section 'building saMBA'. Thus, all extremely rare taxa shouldn't exert an effect on those analyses.

Attending the comments of Reviewers, we have expanded the analysis previously shown in **Figure 3A** (now we show the subsampling results before and after removing rare taxa). In either case, we also performed the analyses using two rarefaction regimes. Please refer to **Suppl. Figure 5** for the explanation of the methods, results and their associated discussion. Briefly, we note that although the effects of removing rare taxa are evident, the effect of rarefying at different levels is negligible.

L235. Why were these analyses carried out at the phylum level and not at the genus level (or both)? If available HMC data is available only at the phylum level place state so. However, it would seem from L287 that such information is available. If the phylum level was chosen so that Suppl. Fig. 1 provides readable results OK, but the analyses in suppl. Fig. 3 (BC) should be carried out at the genus level, as phylum level is extremely coarse-grained.

The analyses resulting in **Suppl. Figure 1** and **Suppl. Figure 3A** of the submitted draft were indeed carried out at the phylum level, as pointed out by the reviewer. Analysis resulting in **Suppl. Figure 3B** and **3C** were carried out at the genus level instead, as suggested by reviewer. To clarify that distinction and to avoid future misunderstandings, we have made that clearer in the methods section and in the text associated with **Suppl. Figure 3** (**Suppl. Figure 4** in our revised manuscript).

L290-303. This seems to be too descriptive in nature. The information contained in panel A conveys the authors' idea that saMBA expands previous knowledge on south American microbiomes. However, the information depicted in panels B and C lacks sufficient context. How do those medians and distributions compare to microbiomes from other parts of the world (processed and analyzed in the same way)?

We have made a new version of **Figure 2** and discussed it in the first major comment made by Reviewer 3 (see above). We hope this fully addressed the potential concerns reviewer had in terms of how microbiomes from South America compare to the rest of the world.

L359. "Additionally, by leveraging saMBA, we generated the most accurate assessment of regional biodiversity to date". The curated database could be used to that end, but there are scarce results on the matter within the manuscript. We believe that after incorporating the suggestions made by Reviewers, our results are the most comprehensive assessment of regional biodiversity, and that they are more accurate than previous efforts. We included more samples and uncovered more unique taxa than the largest global effort (the HMC), as shown in **Figure 2A**. We now have compared the distribution of 2 alpha diversity indices against the regional estimates previously made by the HMC (**Figure 2B**) and against all other world regions in **Figure 2C** and **2D**. Additionally, we have characterised the number of taxa at the level of single countries (**Figure 3A**, **Suppl. Figure 5**) using a randomised subsampling approach with

and without rarefaction to account for sampling depth, and showed how unique the microbiomes of each South American country can be (**Figure 3C**).

L377. And also continuing my previous comment on the matter; a joint analysis of these results against other world regions (processed and analyzed in the same fashion) would be valuable. Similarly, (L378) the authors could have derived Chao1 and Shannon values from the previous compendium to compare using the available count table mentioned in L230. We thank the Reviewer for the suggestions. We have now incorporated this analysis as new panels for **Figure 2**, and we have covered the new results and their discussion in the text. See our response to the first major comment made by reviewer above. Note that Chao1 was changed to Observed genera, as other reviewers had concerns regarding limitations of the former.

L385-387. So saMBA contains samples regardless of the health status? How does this impact the results? How many were from “unhealthy” individuals? How does this relate to the values reported for genera and diversity indices?

Indeed, in saMBA we didn't restrict the inclusion of samples according to what individual studies define as healthy, which is concordant to the approach followed by the HMC. The reasons are theoretical and practical. The definition of what a healthy individual is, in consideration to their gut microbiome, has been debated as non-generalizable across studies. Moreover, it has been discussed that healthy microbiome definitions are usually time-constrained, as gut microbiomes of now-healthy individuals may lead to non-healthy statuses in the near future [see <https://www.nature.com/articles/s41579-024-01107-0>]. Thus, we thought that including all possible compositions of gut microbiomes across the continent would better reflect all different states of the gut microbiomes of South Americans.

Additionally, limitations regarding the extent of metadata sharing across studies made it impossible to identify samples coming from “healthy” or “unhealthy” individuals for every study. This is why we showed the distribution of two alpha diversity indices (instead of single average values). Moreover, when discussing the results we mention that samples in both ends of the distribution come from studies profiling substantially different gut microbiomes. Whereas in the lowest end samples come from children with diarrhoea (and controls), in the highest end samples come from individuals with extremely non-industrialised lifestyles. To further clarify this point, we have now added this to the Limitations section.

L452. “saMBA’ deepens our understanding of the different stable states of the human gut microbiome”. I do not see this statement as related to the content of the manuscript. I agree that saMBA could be used to that end, but the results presented do not particularly relate to that sentence, but are rather directed to exemplify how saMBA outperforms HMC in terms of south American samples.

We thank the reviewer considering future use-cases of saMBA and potentially future resources built to characterised other understudied populations. We agree with the Reviewer that the current work doesn't explore ‘stable states of the human gut microbiome’. Thus, we have rephrased that sentence accordingly.

L455. What do you actually mean when you say that the workflow is compatible with the HMC? Please describe succinctly.

We have extended the assessment of compatibility as reviewers can see in **Suppl. Figure 4** from the revised manuscript. See lines 721-727 from the associated text:

“The Mantel statistic ($r = 0.9813$, $p = 9.99 \times 10^{-5}$, 10,000 permutations) indicated a very strong and statistically significant correlation between the two matrices. This result confirms that our workflow preserves the core structure of microbial community dissimilarities observed in the HMC workflow. Consequently, it supports the compatibility of the two workflows, suggesting that results from saMBA can be integrated with those from the HMC workflow without additional normalization or preprocessing”

Supplementary figure 3C should link (with ellipses, or better with a line) each pair composed of profiles from the same sample analyzed with both pipelines (for genus level analysis). Additionally, I would suggest performing some kind of permutation test or cumulative distribution to better gauge the between-sample dissimilarity (technical) vs among-samples dissimilarities (biological); supplementary figure 3B is too broad scale. E.g. for how many samples does the least dissimilar profile not belong to the same sample but with different processing pipeline?. Or if using per sample empirical cumulative distributions: what is the average and SD of the position that the technical (bioinformatic) replicates occupy along the complete dataset?. This should be done at the ASV or genus level.

We thank the Reviewer for this thoughtful suggestion. Another Reviewer expressed a similar concern and suggested performing a Mantel test to assess the correlation between the pairwise dissimilarity matrix of the HMC and that of saMBA. This test would inform us whether every pair of samples differ from each other in a similar way across workflows. Hence, we have conducted the test and added the results and interpretation to the revised manuscript. See modified text in **Suppl. Figure 4**.

Regarding the suggestions of linking pairs of samples in **Suppl. Fig 3C** (now **Suppl. Figure 4**), we believe that the amount of pairwise comparisons, and therefore, of lines, makes the figure less clear. Additionally, note that the pairwise dissimilarities are already represented as dots in **Suppl. Fig 3B** and assessed through the newly incorporated Matel test. We hope this new quantitative assessment of the strong agreement between workflows is satisfactory to Reviewer 3.

Minor comments:

L42-44. What about host physiology, does it have a role in shaping the gut microbiome?

We thank the reviewer for this comment. We have now added a sentence to the manuscript to cover this aspect.

L49-50. While I follow and agree on the rationale of the sentence, I do not see that the previous observations “indicate that healthy microbiomes and those associated to diseases are different among world regions”. Consider rephrasing.

We thank the reviewer for highlighting the need of refinement in that sentence. We have revised it accordingly.

L140-146. I could not find in the original HMC publication indication that “Projects were discarded if the percentage of non-chimeric sequences is <50% of input sequences” but maybe I am mistaken. On the other hand, what is the rationale for this procedure? I do not fully understand not only the rationale but how it interacts and affects the next filter. Are we talking about ASVs or sequence counts? Are those values related to the total of ASVs sequences or to the total number of sequences belonging to chimeric ASVs. Please clarify in the text.

We thank the reviewer for this important comment. First, we have now changed the word “sequence” to “reads”, to clarify what was done in the analysis. Additionally, we realized the word “discarded” was used interchangeably to describe two different situations: (1) when paired end projects failed to meet the quality criteria but could still be reanalysed as single end, and (2) when the project (single end or paired end reanalysed as single end) was finally discarded from the following analysis. We have made changes to improve the clarity of the text in both ways.

Second, we want to clarify that the rational of including project level quality assessments was explained by authors of the HMC:

“We believe this was the most reliable way to process this data, given the lack of information about project-level sequencing strategies. The merging process in paired-end datasets would be much more effective with more knowledge of study design, particularly in cases where the amplicon length was greater than the read length and the paired-end reads did not overlap. In addition, DADA2 recommends building separate error models for each sequencing run, but only BioProject could be reliably inferred, which means ASV inference at the study level may not capture run-level patterns.”

Hence, suboptimal determination of error rates by the DADA2 algorithm may lead to poor merging and unexpectedly higher levels of chimeric reads, making project-level quality assessment an essential quality check. We have now explicitly stated that in our methods section. Please see lines 167-171.

L154. Just to clarify, by “taxa” you are referring to genera right (not ASVs)?, if so, consider if it is better changing to “genera”.

Yes, we meant genera; it is now changed accordingly.

L154-157: I encourage the authors to include a short discussion on the possible impact of this more stringent threshold. If the applied threshold were to be proportional to the 1000-samples one applied in the case of the HMC, then it should be about 17 or 18 samples. However, a threshold of 100 samples was chosen, making it more than 5 times more stringent. Furthermore, If I understand correctly, the filtering approach appears to exclude approximately 3.8% of the known genera that

were included in the HMC originally according to lines 286-287 — a non-negligible loss of richness, given that 24 additional studies are being included in saMBA.

We thank the reviewer for commenting on this important aspect of the analysis conducted in our research. We want to first clarify an error in the reporting:

In the submitted draft, we reported the number of unique taxa found across all samples of saMBA to be: 2,246. This is corroborated by **Figure 2A**, by summing the number of taxa only found in saMBA, and those found in both saMBA and the HMC (1370 + 876). However, on re-examination there was an error when reporting the filtering process described in the section ‘Building saMBA’ (see methods). We mentioned that we used the same thresholds as the HMC, which we initially did. However, as noted by the reviewer, these thresholds were extremely stringent, as only 296 taxa (~13% of total) across the >2,900 samples passed the filter. Thus, when working on the analysis shown in the submitted manuscript, we used another set of thresholds to be more flexible, despite mistakenly keeping the text describing the original filtering strategy. This can be noted as, in **Figure 3A** from the submitted draft, we state that across all South American samples we found >1,200 different taxa, which would be impossible if we were working with a count table filtered using the approach described in the methods, where only 296 successfully passed all filters.

In this revised version, this error in reporting is fixed. To enhance the clarity and transparency of our work, we have further adjusted the thresholds to filter a proportion of taxa as close as possible to that used in the HMC. Therefore, we have made the following adjustments:

- 5) We have changed the filters steps used to build the ‘clean genus count table’ to match the proportions of reads removed from the count table at each step, as close as possible to what was reported in the HMC article (see methods).
- 6) In addition to stating the thresholds used on each filtering step in the methods section, we added the numbers of taxa and samples ruled out by each step. We added this new information hoping to improve transparency and to make it easier for readers to understand the effects of each filtering step.
- 7) We also updated the code in the GitHub repository to reflect the use of these filters (https://github.com/Benjamin-Valderrama/saMBA-pipeline/blob/main/scripts/misc/cleanup_archive_genus_table.R).
- 8) Finally, we want to clarify that changes in the filtering process would only affect **Figure 3** and **Suppl. Figure 5 in the revised draft** (other figures used the unfiltered count table). In both cases, the figures were made again to account for these changes and due to other comments made by reviewers. Note that the new figures don’t alter the conclusions of our work.

L162- It would seem from the previous sentences that all the filtering was done with the genus-level table; is that so? Is the ASV count table mentioned here the same as was mentioned in L148? Is this ASV count table unfiltered or has it been filtered with the same pipeline described in l153-160? It

would seem from L163-164 (and the repository mentioned) that this is the case, so this is just a minor clarification I need.

We thank the reviewer for this comment. We have now restructured the text to make it clearer. Briefly, we generated ASV-level count tables for each individual study, which were then clustered at the genus level. Then the genus-level count tables of all projects were consolidated into one archive wide genus-level count table. The consolidated ASV-level count table is not filtered or preprocessed. This is described in greater detail in the revised text of the methods section.

L230. “Note that the available HMC count table is from before the preprocessing steps described in the ‘Building saMBA’ section. Therefore, to ensure fair comparisons, the unprocessed saMBA count table at the genus level was used when comparing saMBA and HMC (Figure 2A and 233 Supplementary Figure 3).” Why wasn’t the HMC genus level count table filtered as described in the ‘Building saMBA’ section before the comparative analyses instead? I understand the sentence and those that follow, but feel that I may be missing something with regards to the nature of the HMC count table (see also comment indicating that more information on the nature of HMC data is called for).

The reason to work with the unfiltered table is that we failed to replicate the process used by authors of the HMC to remove taxa. They state that the unfiltered table had 168,464 samples, and they then mentioned that 16,781 samples were removed, as they had less than 10,000 reads. Then, they removed 2018 taxa with less than 1,000 counts across samples, and then they applied a final filter to remove 578 taxa present in less than 100 and samples. Thus, at the end of the filtering process they ended up with 150,721 samples and 1422 taxa.

We downloaded the unfiltered count table using the R package they developed called MicroBioMap (note that the filtered table is not available). It had 168,464 samples, as reported in the original publication. Then, we were able to remove the 16,781 samples with less than 10,000 reads, as they reported. However, when removing taxa with less than 1,000 counts across samples, we removed 2,485 instead of the 2,018 taxa they reported. As we couldn’t replicate their results, we decided to work with the unfiltered count table provided by the HMC, and the unfiltered count table generated in our work for saMBA.

We believe, however, this has an unintended benefit. Users downloading the HMC only have access to the unfiltered genus-level count table. We believe most users wouldn’t probably know so, as the only indication of the type of count table they are downloading comes from the number of samples included in the table. Indeed, authors don’t explicitly tell in the documentation of the R package or their Zenodo repository (<https://zenodo.org/records/15122187>) whether the provided count table is filtered or not. We assumed that a not neglectable proportion of users will work directly with the table without reading the HMC paper in detail, thus they wouldn’t know if the table is filtered or not.

Note, however, that to avoid confusion among potential saMBA users, we have provided both types of tables in our Zenodo repository. Moreover, we have expanded now the details of our filtering process in the methods section.

L322-323. What do you mean? In what sense? Consider expanding the argument just a little bit (or remove the sentence).

We used that sentence to introduce the analysis described in the following paragraph (**Figures 3B and 3C**). We think that probably the use of ‘Next’ in the following paragraph can be confusing. Thus, we have changed it to ‘Hence’ to make it clear that the analysis described follows from the previous paragraph.

L366-368. Why restrict the space to continents? The argument feels unsupported and related to what the authors present and not a particular scientific question of interest. You could argue as the authors did early in the manuscript that “As geographic location represents an ensemble of genetic, environmental and cultural factors,” (If so consider reformulating the statement on these grounds). However, wouldn’t it be better to limit the search space to meaningful populations with different genetic, environmental and cultural factors? The authors actually delve on this on L414-L432. We fully agree with the logic of this comment, and as Reviewers 1 and 3 highlighted, we are aware that even more restrictive search spaces are equally—or potentially more—meaningful than continents for certain scientific questions. This is especially relevant where environmental and cultural factors occur in gradients within close geographic proximity, as we highlighted latter in the text, when we describe contrasting lifestyles in Brazilian cities and communities within the Amazon. Moreover, it was further expanded in the text of the **Suppl. Figure 6**.

We thank the reviewer for highlighting this sentence, as it was indeed not intended to convey the idea that continents are the ideal search space. Instead, our intent was to suggest that performing the analysis at the level of global regions, as used in the HMC, is probably too coarse. We showed that in our **Suppl. Figure 1**, where we identified a visible distinction between Central and South America, despite being unified into the category “Latin America and the Caribbean” in the HMC. This information was used then to justify the restriction of the search space to South America, which is a finer level of resolution than Latin America and the Caribbean, but still with potential to be of interest to a broader community of researchers and readers.

To avoid potential misinterpretations, we changed the phrasing to remove emphasis on the word continent.

L387-389. This is expected, I would not consider the use of the term “validation” in this scenario. We have removed the sentence from the manuscript. The intention behind it was to highlight that preprocessing parameters used to build the HMC, and then saMBA, allowed to replicate some of the biologically relevant results of the original publications, even if they weren’t optimized for the analysis of any particular study included. Thus, the word validation was used for the preprocessing parameters of the pipeline, not the biological findings.

In Figure 3, I suggest including the actual numerical range of the represented LRI values, at least in the caption or the main text. This would help the reader interpret the scale and relevance of the observed differences.

Thank you for the suggestion. We have now added the LRI values for the 2 highest and lowest countries in the main text.

Reviewer #3 (Remarks on code availability):

The code and datasets included in the repositories are well presented and clear. In fact, the authors' efforts towards transparency and open science are commendable, including the public availability of the pipeline, the extensive documentation, and the invitation to external researchers to directly share new studies for their inclusion in the database. Nonetheless, we spent a considerable amount of time trying to install the saMBA pipeline on Linux, but ultimately we were unsuccessful. We're still unsure whether the issues stemmed from some incompatibility on our end, but we tried several troubleshooting steps: installing micromamba instead of using conda, removing conda's activation steps from the ~/.bashrc file altogether, resetting the PYTHONPATH to an empty string, removing some possibly conflicting packages from our system, and even simplifying the .yml file by removing version constraints and all R packages. None of these attempts resolved the problem. Perhaps the pipeline runs smoothly on a freshly installed Linux system, but in our case we were unable to get it working. We also tried running the script without installing anything, but without fastq-dl, which we also failed to install, we weren't even able to download the test files. As other researchers could face similar issues, a possible option would be to provide a Docker container, which is fully self-contained and would ensure reproducibility and usability.

We thank Reviewer 3 for the positive comment on our efforts to make the code open to the community. We transitioned to Docker to ensure full reproducibility, and to facilitate the deployment of the workflow by third parties. The file to build the container is now shared in GitHub. The built image can be accessed from Docker Hub and instructions to download and deploy the workflow in Docker can be found in the workflow documentation and the README file, where Docker is the recommended option.

Reviewer #4 (Remarks to the Author):

Reviewer #5 (Remarks to the Author):

I co-reviewed this manuscript with one of the reviewers who provided the listed reports. This is part

of the Nature Communications initiative to facilitate training in peer review and to provide appropriate recognition for Early Career Researchers who co-review manuscripts.

We thank co-reviewers 4 and 5 for reading their work and providing comments to improve it.

We thank all the Reviewers for the positive reception of our work especially in terms of its relevance and for acknowledging our efforts in making this article (and the resources associated) open to the community.

Below we outline our point-by-point response to the remaining comments.

REVIEWERS' COMMENTS

Reviewer #1 (Remarks to the Author):

The authors have performed thorough revisions in response to the feedback from their paper's many reviewers, including multiple new analyses and figure panels. The careful edits (and patient descriptions in the response document) were very useful. The paper is useful and interesting, and I have no further feedback that should preclude publication.

Reviewer #1 (Remarks on code availability):

The analysis scripts for the paper are straightforward and well-commented. The repositories, plus associated wikis and website, are exemplary.

We thank Reviewer #1 for their appreciation of the relevance of our work, as well as the effort we put in conducting the new analyses and preparing new figures. We also thank the recognition made regarding our efforts to make all associated resources open to the community.

Reviewer #2 (Remarks to the Author):

We thank the authors for the changes made. We consider that both the manuscript and the pipeline have improved significantly, and that our initial comments have been adequately addressed.

We thank Reviewer #2 for the thoughtful comments made in the previous round of reviews. We truly believe that made our work improve.

There are two points we would still like to highlight:

1) The filtering of the studies (and its effects). It is surprising that the initial ENA search identified 255 studies, yet only 23 passed the filter—less than 10%. The remaining studies were imported from the HMC, which is also unexpected given that the HMC itself is based on an ENA search. It is therefore unclear why these studies did not appear in the initial search. It may be worth discussing what prevents the majority of studies from being usable in a meta-analysis. Is it the lack of metadata? Errors or inconsistencies? We believe it would be valuable for the manuscript to highlight this issue, as it emphasizes the need for improved study reusability.

We thank Reviewer #2 for the comments aimed at clarifying the methods used to shortlist the projects included. As correctly highlighted, less than 10% of the originally screened projects were included, and the reason is mentioned in the manuscript (see lines 101-102):

“Broad search terms without restriction of publication year were used to avoid early exclusion of potentially relevant bioprojects”.

Consequently, many microbiome sequencing projects matching the words used in our broad search term were completely unrelated to our research question. Take for instance the results of searching by “(microbiome OR microbiota) AND (Chile OR chile)”, one of the examples used in the manuscript.

While most projects profiled environmental microbial communities, only two projects from the list of initial results may be relevant to the goals of saMBA. PRJNA446042 and PRJEB16755. The first exclusively sequenced samples from a body site different from the gut, which is stated as one exclusion criteria in the manuscript (please see lines 115–125). Thus, only PRJEB16755 could be included (and it was).

Regarding as to why some studies included in saMBA were not included in the HMC, we can't offer a definite answer, as exact details of how their screening was performed are not available to us. However, we speculate that it may be due to the automated screening of the databases performed in the HMC. That method overly relies on the accuracy of the metadata provided by the users uploading the raw data to the databases. Indeed, the HMC highlights that as a limitation of their work when stating that their screening process led them, for instance, to the inclusion of vaginal microbiomes that were incorrectly identified as intestinal microbiome samples in the INSDC (see their methods section).

We believe that our approach of manually curating the datasets included can help overcome that limitation, as stated in lines 323-324: *"This highlights the need for manual curation of studies metadata for more accurate assessments of regional microbiomes"*.

We are aware, however, that this approach is potentially limited to searches within narrower geographic limits, making it hard to scale the search to the extent the HMC worked with. That's why we believe this work, and the resources generated, is also a good complement to the HMC.

2) Although some actions have already been taken in the discussion, we still feel they fall short. In particular, when reviewing the alpha and beta diversity analyses, the figures and data are typically labeled by country. This presentation can easily lead readers to assume that the results are representative of entire countries, despite clear biases in the underlying data. As mentioned in the manuscript, the high diversity observed in Peru is likely due to the fact that most samples are from the Amazon region, while the low diversity in Argentina likely reflects that all studies were conducted in industrialized settings. We suggest explicitly showing, in the figures, the number of studies and samples per country to help readers better assess the representativeness of the results.

We thank Review #2 for highlighting our efforts in trying to avoid misleading readers. We agree with the reviewer that sampling is not evenly distributed across territories. This comment was made in the previous round of reviews, to which we added lines 513-516 in the previously submitted revised manuscript.

We also agree with the current suggestion about explicitly showing in the figures the number of studies and samples per country. This information was also available to readers in Figures 1B and 1C in the manuscript submitted.

One minor comment:

1) Just out of curiosity: if "South America" is always capitalized, why is the "SA" in saMBA lowercase?

Although there wasn't any reason originally, the lowercase differentiates our work from other computational analyses called SAMBA.

Reviewer #2 (Remarks on code availability):

We thank the authors for adding an additional layer of reproducibility and containerization to the pipeline using Docker. While this introduces some complexity for users—particularly when verifying the Docker

image and running it interactively—we recognize this as a valuable addition that functions well overall. However, we recommend reviewing the `wget` commands used for file downloads, as they may be slower than expected or prone to failure. This could be related to the limited resources typically allocated to the default Docker image, which may differ from the environment used during development.

We thank reviewer #2 for recognizing our efforts towards reproducibility. We agree that using Docker adds an extra layer of friction for some users, but we believe that its benefits in terms of reproducibility outweigh potential downsides.

Regarding the rare instances where the download of data failed, we opened an issue on the GitHub repository of the software used to download data in February this year. On our side, we are open to evaluate using different software for that purpose if users report problems.

Reviewer #3 (Remarks to the Author):

The rebuttal letter and modifications to the manuscript adequately address all my concerns and comments.

Reviewer #3 (Remarks on code availability):

We appreciate the decision to follow our suggestion of a Docker container. As well as the complete rewrite of the documentation to reflect this change — it's very thorough.

The installation is extremely simple now and takes just a few minutes. We were able to run the demo completely, move the files to our machine and look through them. We were also able to set up a docker volume and rerun the demo with both the input and output folders mapped to our local machine. Additionally, the pipeline seems to be executable as well using the `input_samba_v0.1.0.txt` available on Zenodo. We did not run it fully, however, because of time and disk space constraints.

We thank Reviewer #3 for recognizing our efforts towards transparency and reproducibility, and for the positive comments about documenting our workflow. On a side note, mounting the volume and running the analysis using that file as input is exactly how we built the current version of saMBA that's available in Zenodo.

The scripts on the saMBA-article repository seem to be thorough, and comments on the code itself are generally clear. It's unfortunate that the files on both this second repository and Zenodo do not include all the files needed to run the scripts without any modifications — the available metadata files seem to be missing or in a different format than needed. Still, we were able to replicate Figures 2b and c from the count table — so at least there's some confirmation that the figures can be reproduced from the count table.

We thank Reviewer #3 for this observation. The files couldn't be uploaded to GitHub as they exceed the maximum size of the platform. Therefore, we uploaded them to Zenodo. It is true that originally the metadata file was missing, but that mistake was corrected in the latest release (v1.0.1), published on Zenodo on the 15th of June along with the `changelog.md` file that keep tracks of that and all future changes: <https://zenodo.org/records/15663639>.

Reviewer #4 (Remarks to the Author):

Reviewer #4 (Remarks on code availability):

Reviewer #5 (Remarks to the Author):

We thank co-reviewers #4 and #5 for reading their work and providing comments to improve it.